

# Seasonality and spatial variability of dynamic precipitation controls on the Tibetan Plateau

Julia Curio[1,2] and Dieter Scherer[1]

[1]Chair of Climatology, Technische Universität Berlin, Berlin, Germany
[2]Department of Meteorology, University of Reading, Reading, UK

*Correspondence to*: J. Curio (julia.curio@tu-berlin.de)

**Abstract.** The Tibetan Plateau (TP) is the origin of many large Asian rivers, which provide moisture for large regions in South and East Asia. Therefore, the water cycle on the TP and adjacent high mountain ranges and especially the precipitation distribution plays an important role for the water availability for billions of people in the regions downstream of the TP. Based on the High Asia Refined analysis (HAR) we analyse the influence of dynamic factors on precipitation, enhancing or suppressing precipitation development. We selected six precipitation controls, the horizontal and vertical wind speed at 300 hPa and in about two kilometres above ground, the atmospheric water transport, and the planetary boundary layer height. Focus of the study are the seasonality and the spatial variability of these precipitation controls and their dominant patterns. The results show that precipitation controls have different effects on precipitation in different regions and seasons. This depends mainly on the type of precipitation, convective or frontal/cyclonic precipitation. Additionally, the study reveals that the mid-latitude westerlies have a high impact on precipitation distribution on the TP and its surrounding year-round.

## 1 Introduction

The Tibetan Plateau, often called the "world water tower" (Xu et al., 2008), is the origin of many rivers in high Asia, which provide water for billions of people in the downstream regions in South and East Asia. The TP is located in the transition zone between the impact of the mid-latitude westerlies (Schiemann et al., 2009) and the Indian and East Asian summer monsoon systems (Webster et al., 1998) and shapes the hydro-climate of downstream regions by its influence on large-scale circulation (Hahn and Manabe, 1975), caused by its large extent and height.

Since precipitation is a key feature of the water cycle of High Asia, it is important to analyse the factors controlling precipitation. This may help to estimate the impact of future climate change on precipitation development and amounts, since if a specific control in a specific region and time is identified, one would be aware that a strengthen or weakening of this precipitation control has distinct impact on precipitation development.

Previous studies stated that precipitation on the TP is controlled by the westerlies in winter and the Indian and East Asian summer monsoon in summer (e.g. Hren et al., 2009; Tian et al., 2007; Yang et al., 2014). This assumption is based only on precipitation timing, but Curio et al. (2015) and Mölg et al. (2013) already showed that the mid-latitude westerlies have also an impact on summer precipitation.

There are many studies on the influence of atmospheric circulation modes, e.g. the North Atlantic Oscillation, the Arctic Oscillation, and El Nino/Southern Oscillation, on climate and precipitation in high Asia and on the monsoon systems (e.g. Bothe et al., 2009; Liu et al., 2015; Liu and Yin, 2001; Rüthrich et al., 2015). Less attention is drawn to the underlying processes controlling precipitation development on the TP and the surrounding high mountain ranges.





Using the High Asia Refined analysis (HAR) (Maussion et al., 2014), which is the result of dynamical downscaling of a gridded global data set and covers more than thirteen years, on a daily basis, gives us the opportunity to analyse the data process based.

Starting point of this study is the precipitation seasonality on the TP and the question which factors control precipitation on the TP and the surrounding high mountain ranges. Therefore, we want to examine which precipitation controls has where and when an influence on precipitation development and how strong it is. Aim of this study is to describe the correlation between controls and precipitation spatially and temporally differentiated. We selected six variables as dynamic precipitation controls, horizontal and vertical wind speed at 300 hPa and in about two kilometres above ground, the vertically integrated atmospheric water transport and the height of the planetary boundary layer. It is known that these factors have an influence on precipitation development, but on the basis of coarse resolution datasets it was not possible to analyse the relations spatially and temporally differentiated like it is now with the High Asia Refined analysis (HAR). We do not claim completeness for the list of precipitation controls, but assume that these six factors belong to the most important dynamic precipitation controls. It is important to keep in mind, that the precipitation controls are not independent from each other and can have combined effects on precipitation development or neglect each other, this will not be in the focus of the current study.

The selection of precipitation controls relies on well studied relations of these factors with precipitation (e.g. Back and Bretherton, 2005; Garreaud, 2007; Shinker et al., 2006) but these controls were not investigated at high spatial and temporal resolution in high mountain Asia until now.

The different precipitation controls have effects on different spatial scales. While the horizontal and vertical wind speed at 300 hPa are large-scale controls, their counter parts in the boundary layer and the PBLH are active on the meso-scale. The AWT connects the large- with the meso-scale because it is effective on both scales and across large distances. In the following we will shortly introduce the selected precipitation controls and their possible impacts on precipitation development.

The horizontal wind speed at the 300 hPa level has two main effects on precipitation development. High WS300 can inhibit or cut off deep convection and thus supress precipitation development. In this case only shallow convection can form which does not lead to considerable precipitation amounts. Mölg et al. (2009) showed that convective precipitation events on tropical mountain summits correspond with low horizontal wind speeds. On the other hand, high wind speeds enhance moisture advection and thereby strengthen frontal/cyclonic precipitation and also orographic precipitation. At the 300 hPa level we are already in a height where the westerly jet occurs, which strength and location influences the hydro-climate of the TP and central Asia (e.g. Schiemann et al., 2009). Especially the precipitation seasonality in the north-western parts of the study region is related to the position of the jet (Schiemann et al., 2008).

Higher horizontal wind speeds in the planetary boundary layer can hinder convection and advect air masses/moisture like its counterpart at 300 hPa, but also can enhance evapotranspiration from the surface (Back & Bretherton, 2005) and therefore convection, depending on the available moisture at the surface. This process is interesting mostly during the warm half of the year, when surface moisture from local sources like lakes, soil moisture, the active layer of permafrost, snow and glacier melt is available. Moisture recycling plays an important role for precipitation on the TP (e.g. Araguás-Araguás et al., 1998; Trenberth, 1999), on average more than 60% of moisture needed for precipitation falling on the inner TP are provided by the TP itself (Curio et al., 2015).

Back and Bretherton (2005) found that there is a relation between near surface wind speed and precipitation in the Pacific ITCZ. Higher wind speeds can intensify the evapotranspiration rate and therefore the development of deep convection.





The negative effect of high horizontal wind speeds on precipitation plays an important role in regions and seasons which are dominated by convective precipitation e.g. the Tibetan Plateau in summer (Maussion et al. 2014). While the positive effect occurs in regions with mainly frontal/cyclonic precipitation because the moisture transport towards these regions is enhanced by the strengthened atmospheric flow.

Garreaud (2007) pointed out that stronger than normal low level westerlies lead to more precipitation on the luv side of meridionally orientated mountain ranges (orographic precipitation), while high wind speeds at mountain tops leads to rather dry conditions because of intensified downdrafts. This process is called rain shadow effect. But he also shows this could lead to more precipitation on the lee side because more cloud particles are advected and disturbances can overcome the topographic barrier with the help of higher wind speeds, which would lead to more frontal or cyclonic precipitation. This case already shows that it depends on many factors which influence each of the dynamic controls has on precipitation and that the of influence highly varies regarding time and space.

The influence of the vertical wind speed on precipitation development depends on the direction of the vertical wind, whether it is an updraft or a downdraft. Updrafts have a positive impact on precipitation because they can boost or enhance convection and are a key element for orographic precipitation on the luv side of mountain ranges, while downdrafts (e.g. on the lee side of mountain ranges) and subsidence lead to inhibition of convection and cloud dispersal. The expectation is that we have mainly positive correlations of vertical wind with precipitation, high upward winds cause higher precipitation and higher downward winds cause less precipitation.

Atmospheric water transport (AWT) is not only a dynamic precipitation control because it is a product of atmospheric moisture content and wind speed (and direction). We assume that there is always a positive correlation between AWT and precipitation, because AWT delivers moisture for precipitation from remote sources to the TP (regions where precipitation occurs). But high AWT does not automatically lead to precipitation development, what was for example shown by Curio et al. (2015) for the Qaidam Basin where the prevailing atmospheric subsidence inhibits convection.

The height of the planetary boundary layer (PBLH) shows the turbulent mixing of the lower atmosphere. It is influenced by solar forcing, turbulent processes, convective activity (Yang et al., 2004), moisture content and temperature of surface and air, and wind speed. Gentine et al. (2013) point out that dry soil surface accelerates the growth of the PBL. The TP has one of the highest PBLs in the world, due to its high altitude and strong solar forcing, but the TP PBL is not fully understood yet because there are only few observations. During daytime in spring and summer the PBL is higher than in the other seasons in this region (Patil et al., 2013) and can exceed heights of 3000 m or more (Ma et al., 2009; Yang et al., 2004) in some regions on the TP due to strong solar forcing and convective activity. The horizontal wind speed influences the PBLH by facilitating the mixing of the atmosphere and the entrainment of air from above the PBL. The PBLH is not a pure dynamic precipitation control, but entrainment as a dynamic factor has great impact on PBLH (Medeiros et al., 2005; Gentine et al., 2013). Additionally, in the HAR the PBLH is determined dynamically by using the Mellor–Yamada–Janjic turbulent kinetic energy scheme (Janjic, 2002) for PBL parameterisation.

The objectives of the current study are two-fold:

    i. analyse the impact of the selected dynamic controls on precipitation development spatially and temporally differentiated, where and when has which precipitation control a how strong influence,



ii. examine if precipitation controls have always and everywhere in the study region the same impact or whether there are opposing effects active in different sub-regions,

iii. and gain a better understanding of the role of the mid-latitude westerlies and the summer monsoon systems for precipitation distribution on the TP.

Here we first present the precipitation seasonality on the TP and adjacent mountain ranges based on the HAR using a cluster analysis. Focus of the study is the period 2001-2013. Afterwards we analyse the correlations between the six selected dynamic precipitation controls and precipitation, regarding seasonality and spatial variability. A principal component analysis of this correlations is then used to detect the dominant patterns.

## 2 Data and Methods

### 2.1 The HAR data set

This study is based on the High Asia Refined analysis (HAR). The HAR is generated using the advanced research version of the Weather and Research Forecasting model (WRF-ARW, Skamarock and Klemp, 2008) version 3.3.1 for dynamically downscaling of the Operational
Model Global Tropospheric Analyses (final analyses, FNL; data set ds083.2), a global gridded data set. The HAR dataset, its modelling, forcing, and re-initialization strategies, and the validation of the results are described in detail by  Maussion et al., (2011, 2014). The first domain of the HAR encompasses most parts of south-central Asia with a spatial resolution of 30 km and temporal resolution of 3 h (HAR30). High Asia and the Tibetan Plateau are the focus
of a second nested domain with a spatial resolution of 10km and temporal resolution of 1 h (HAR10). A map of the HAR10 domain and its location in the parent domain HAR30 are shown in Fig. 1. In this study we analyse the processes on the TP and the surrounding high mountain ranges and therefore use the HAR10 data only. The calculation of the vertically integrated atmospheric water transport can be found in Curio et al. (2015). For this study the HAR data
are used on a daily basis.

### 2.2. Methods

For the analyses in this study we selected the years 2001-2013 as study period and use daily HAR data. The data set is then stratified, because we are only interested in precipitation days, this is done month-wise. For example, all July analysis depend on the data basis of each July
day during 2001-2013, these are 31*13 days, total 403 days. Therefor all analyses are conducted for precipitation days only. Precipitation days for a grid point are defined as days with a mean daily precipitation rate of 0.1 mm. This value is calculated as the mean precipitation rate for all grid points within an area of 15*15 grid points around the grid point. The time mask for precipitation days is therefore applied on the variables used as precipitation controls. The
number of precipitation days can vary distinctly between regions and seasons. Each of the precipitation controls is then correlated with precipitation using the Spearman rank correlation, this is done month-wise, for all precipitation days in a specific month during the study period and for each grid point in the HAR10 data set. Only significant correlations, with a level of significance of 0.05, are plotted. Following this, a principal component analysis (PCA) was
performed for the correlations between controls and precipitation to detect the dominant patterns.
Due to the use of correlations we avoid problems regarding the exact precipitation rates/amounts falling on the TP, which are hard to measure and to model. The starting point of
this study is the precipitation seasonality on the TP which was analysed by Maussion et al.





(2014) using the k-means clustering method (e.g. Wilks, 1995) and will be deepened in the current study. The percentages of monthly contribution to annual precipitation and not the precipitation amounts were used to define seven classes with different precipitation seasonality. This has the advantage that in an area like high Asia where the differences in precipitation amounts vary strongly between regions and seasons, the regions were made comparable by this method. We conducted the cluster analysis with other numbers of classes, but the chosen number of seven classes led to the best ratio between coherent patterns and sufficient distinction between classes.

## 3 Results

### 3.1 Precipitation seasonality

Figure 2 shows the seven defined classes of precipitation regime, and the mean annual cycle of monthly precipitation percentage in each class. It is clearly visible that the TP and the surrounding mountain ranges show a spatial variability regarding precipitation seasonality. A class with precipitation maximum in summer and minimum in winter is dominant on the central TP and south of the Himalayas in India, Nepal and Pakistan (blue). This class is the monsoonal precipitation class because the precipitation percentage increases from June on, the onset period of the Indian summer monsoon. During July and August more than 50% of the annual precipitation are falling in this regions, while from October to May the monthly precipitation amount is below 5 % of the annual precipitation. The yellow class has a much broader and less intensive summer precipitation maximum and the increasing and decreasing proceed with similar rates. The annual cycle of precipitation in this class is determined by the seasonal cycle of solar forcing and therefore convective activity. The class laying between the so called monsoonal and convective classes represents a region of mixing between monsoonal and convective classes, there we can have monsoonal precipitation and/or only solar forced convective precipitation (green). This class represents a transition zone where the monsoon can have influence but is not dominant. The naming convention does not mean that the monsoonal class precipitation is not of convective nature, but it should emphasize that the precipitation development is influenced by the monsoon, which is associated with the advection of tropical air masses. The monsoonal class is a subset of the convective class. That there is a forcing difference between the two classes is clearly visible in the timing and strength of the precipitation maximum which is higher and later in the monsoonal class and starts during the Indian Summer Monsoon onset period and persists only for a shorter time period. The monsoonal class is divided in a northern and southern part by the Himalayas.

The grey-blue class experiences it precipitation maximum in winter and is dominated by the influence of the mid-latitude westerlies. This class occurs mainly in the Pamir/Karakoram/western Himalayas (PKwH) region, as one coherent pattern, and additionally in the eastern part of the Tarim Basin. The precipitation maximum is much lower, but the minimum values are higher than in the classes dominated by summer precipitation, which shows that the intra-annual variability is lower, the values vary between 15% and 5%, but are never below 5% in the mean. This could lead to the assumption that the influence of the mid-latitude westerlies is more constant year-round, while the monsoon has a stronger but temporally more limited influence on precipitation on the TP.

The light blue class exhibits a more even distributed precipitation seasonality, with spring and summer precipitation and a minimum in November. This class occurs as a frame for the grey-blue cluster, in the south-eastern TP and in the eastern Tarim Basin. It is interesting that the region in the south east of the TP, where the Brahmaputra Channel enters the TP belongs to a different class than the surroundings which are divided between the three convective dominated



classes, blue, yellow and green. Maybe there is stronger influence by AWT, which would make this region more similar to the surrounding of PKwH region regarding the factors controlling precipitation development.

The remaining two classes, red and purple, almost only appear in the Tarim Basin. They are both characterized by a spring, early summer precipitation regime, but also show some differences. The amount of precipitation in the purple class increases sharply from March to May/June were the maximum with a value of slightly above 20% occurs. The decrease is even sharper so that in July already only 6 % of the annual precipitation occur. The annual cycle of the red class seems to be a bit delayed against the purple one and the period of maximum precipitation is only one-month long. In spring the values are lower but higher in late summer and autumn, while they are almost the same during winter.

The high mountain ranges of the Himalayas and Kunlun and Qilian Shan exhibit a complex structure of different classes on relatively short distances and therefore exhibit no coherent patterns. This holds true also for the border area between the Karakoram and the Tarim Basin. The mountainous region of the Pamir and Karakoram are represented by only one class, which implies that this region is mainly influenced by one atmospheric forcing (mid-latitude westerlies) or that different controls have the same impact in this region. The other mountain ranges may lie in regions where an interplay of different controls occurs, and the temporal and spatial variability is larger on smaller scales.

## 3.2 Correlation of dynamic controls and precipitation

We only show the months with the most prominent features. Figures with the correlation of each dynamic control with precipitation for all months are provided in the supplemental material for completeness.

### 3.2.1 Horizontal wind speed

a) Horizontal wind speed at 300 hPa (WS300)

The correlations between WS300 and precipitation are provided in Figure 3 for selected months. There are high positive correlations in winter in the PMwH region. This is the time of the year when the majority of precipitation falls in this region (e.g. Curio et al., 2015; Maussion et al., 2014). The precipitation in this region is mainly cyclonic/frontal precipitation, which benefits from enhanced moisture supply due to higher wind speeds. On the TP we have only negative correlations in the southern and eastern parts in winter, while this area enlarges over spring and covers mainly the whole central and north-eastern TP and large parts of the central Himalayas with the highest negative correlations in the central TP. Reason for this is the dominance of convective precipitation in this region (e.g. Maussion et al., 2014). Higher wind speeds inhibit deep convection and therefore have a negative effect on precipitation development. This also explains why there are almost everywhere negative correlations in summer. The region of negative summer correlations has almost the same shape like the convective and monsoonal precipitation classes (Fig. 2) put together, so the impact of WS300 explains some of the precipitation clusters. In spring there is a second region with positive correlations which lays north of the TP in the Tarim Basin and the bordering Kunlun and Qilian Shan. This area is reached by the northern branch of the mid-latitude westerlies which delivers moisture for precipitation. The area around the Brahmaputra Channel exhibits slightly positive correlations, due to enhanced moisture transport to by higher wind speeds. The region of high




positive winter correlations exhibits no or slightly positive correlations only in a small area in summer, because there is almost no precipitation (0-5% of annual precipitation) during summer months in the Pamir/Karakoram region (Maussion et al., 2014).

b) Horizontal wind speed at model level 10 (WS10)

Figure 4 shows, that the correlations between WS10 and precipitation are positive in the PMwH region in winter (February) like they are for WS300, but the region is larger, especially the region with correlations > 0.6. Reason for the positive correlations is again the enhanced moisture supply due to higher wind speeds and therefore also more orographic precipitation. The positive correlations in the Brahmaputra Channel region (and south of it) are more
pronounced here and the structure of the Brahmaputra Channel itself is clearly visible. The moisture supply is enhanced due to strengthened winds from the south, bringing moisture from the Indian Ocean to the Himalayas. There are high positive correlations between the WS10 and precipitation in summer, while the correlations with WS300 are negative in summer. This is because of the fact that the wind speed in the boundary layer can enhance evapotranspiration
from the surface (e.g. lakes; snow, glacier and permafrost melt), which leads to more moisture in the lower atmosphere available for precipitation. This correlation again emphasizes the importance of moisture recycling on the TP. In winter the correlations on the TP are mainly negative because the effect of enhanced evapotranspiration is not active due to the fact that all potential moisture sources are frozen during this time of the year.

3.2.2 Vertical wind speed

The correlations between vertical wind speed in 300 hPa (Fig. 5) and model level 10 (not shown because of high similarity; see supplemental material) and precipitation are mainly positive due to the positive effect of ascending air motion on precipitation development, like assumed,
especially in summer when most of the precipitation is convective. Therefor, the correlations are higher in summer than in winter when almost no precipitation falls on the TP, although the mean vertical wind speeds (up- and downdrafts) are higher in winter than in summer (Fig. 6). This shows that higher values of one precipitation control alone do not necessarily lead to higher correlations and therefore more precipitation, but that usually other conditions favourable for
precipitation development have to occur. This points out/emphasizes the assumption that precipitation development is mostly caused by combined effects of precipitation controls.
The high negative correlations in the high mountain ranges of the Pamir and Karakorum in winter could be an aggregation problem, due to the use of daily means. This would be the case if we, for example, have a day with strong downdrafts in the mean, but during a shorter time of
the day we have updrafts leading to precipitation, which are not seen in the mean for the day. Other possible reasons for negative correlations could be large systems of frontal/cyclonic precipitation, which can still rain on the lee side of the mountains, where as a consequence of the flow over the mountain ranges already downdrafts occur. This would lead to the simultaneous occurrence of downdrafts and precipitation, and therefore negative correlations.

3.2.3 Atmospheric water transport (AWT)

Figure 7 provides the correlations between AWT and precipitation. The entire study region is dominated by positive correlations during the year (Fig. 7), like we assumed before. There are only few regions and months where negative correlations occur, in the Tarim Basin during
winter, at the central Himalayas (and the southern slopes) during summer. The highest positive correlations occur in winter and early spring in the PKwH region, the correlation coefficient r exceeds 0.8 in large areas. This is the time of the year where the precipitation maximum occurs





in this region (Fig. 2). The annual portion of convective precipitation in this region is below 10% (Maussion et al., 2014), the region exhibits orographic precipitation triggered by the advection of moist air masses with the westerly flow and westerly disturbances (Cannon et al., 2014). Therefor higher moisture supply leads naturally to more precipitation. The positive correlations in the western parts of the study region persist during the course of the year, even if their extent and strength varies. From this centre the positive correlations expand to the TP, the whole Himalayan arc, and along the northern border of the TP (Kunlun and Qilian Shan), these seem to be the moisture supply routes along the branches of the mid-latitude westerlies.

A second centre of positive correlation which is persistent is found in the extreme southeast of TP, the region where the Brahmaputra Channel enters the TP, this region exhibits also less convective precipitation on annual scale (Maussion et al., 2014), while it is surrounded by regions with high convective precipitation rates. This explains the fact that this region belongs to a different precipitation seasonality class than its surroundings (Fig. 2). In January and February, the Himalayan arc exhibits mostly positive correlations and connects the area in the western parts of the domain with the other positive centre in the south-eastern TP around the Brahmaputra channel.

The spatial minimum of the positive correlations occurs in May, whereas the correlations are very high (r > 0.6) in the western and south-eastern centres. In May the central TP exhibits no significant correlations which may be caused by the fact that the AWT is very low on the TP in May (Curio et al., 2015). From now on the area with positive correlations enlarges but the strength of correlation decreases.

The minimum values of the positive correlations occur in July and August. But now we have positive correlations on the central Tibetan Plateau, but they are not as high as in the western and south-eastern parts of the domain during winter and spring. This shows that the precipitation during this time of the year is mostly convective whereby the advection of moist air masses is less important. In large areas of the domain the precipitation maximum occurs in July and August, ~30% of the annual precipitation are falling on the central and southern TP during this time (Maussion et al., 2014).

The regions where the highest positive correlations occur are the regions where the precipitation maximum occurs during winter, matching the grey-blue precipitation seasonality class (Fig. 2).

### 3.2.4 Planetary boundary layer height (PBLH)

In Winter the correlation between PBLH and precipitation (Fig. 8) are mainly positive, especially in the PKwH Himalaya region, forming a coherent pattern, and along the northern and southern border of the TP, with some interruptions, and in the eastern TP around the Brahmaputra Channel. PBLH is a precipitation control which is itself controlled by horizontal wind speed, among others. The positive correlations in winter in the Pamir/Karakoram region can be explained by higher horizontal wind speeds and thereby more moisture advection leading to more precipitation, like described earlier. Higher horizontal wind speeds lead to higher entrainment rates at the top of the PBL whereby the depth of the PBL increases.

During the year this picture changes and negative correlations become dominant. The negative correlations occur in regions and seasons dominated by convection and consequently convective precipitation. Higher PBLH are caused by higher horizontal wind speeds which have a negative effect on precipitation by cutting off deep convection, as previously pointed out in this study. In May we see positive correlations left only in the centre of the Pamir/Karakoram region and in the eastern TP, while in summer just small and scattered areas with positive correlations occur. Most of these areas on the central TP match with the large lakes, e.g. Nam Co, Serling Co, and Tangra Yumco.





To understand these contrary correlations, compared with the direct surroundings of the lakes, we have to look at the PBLH itself. Figure 9 shows the averaged daily mean and daily maximum PBLH for May as an example. The lowest PBLHs (mean and maximum) in the domain are found at the high mountain ranges bordering the TP, because of the high altitudes and low temperatures, and above the large lakes on the central TP. Reason for this seems to be the fact that during daytime when the air temperature above land is higher than above the lakes, the air above the land starts to ascend and there is strong convective activity which is amplified at the slopes of the surrounding mountain ranges (due to higher solar insulation). Recirculation of the ascended air leads to subsidence of air above the lakes which lowers the PBLH and supresses precipitation. Suppression of precipitation by subsidence was already described in Curio et al. (2015) for larger regions like the Qaidam Basin and the north-western parts of India and Pakistan.

The positive correlations above the lakes, mean that higher PBLH leads to more precipitation in this area. These higher PBLH occur when the wind speeds are higher, which leads to more entrainment, which causes a deepening of the PBL above the lakes. This effect slightly equalizes the PBLHs above the lakes and the surrounding land and the differences become smaller. Due to the increase of the PBLH, the prevailing subsidence above the lakes weakens and precipitation development becomes possible.

## 3.3. Principal components

We described the spatio-temporal variability of correlation between controls and precipitation for the TP and surrounding high mountain ranges. To detect the leading or dominant patterns in this relations and to find out in which time of the year a control is most efficient, we conducted a principal component analysis (PCA) for each of the correlation sets. Because we are interested in the dominant modes we only analyse the first two principal components (PCs), which explain most of the variance of the data set. The patterns in the higher PCs, which also can explain important processes, will be subject of a further study. For completeness plots for all PCs can be found in the supplemental material of this study. Explained variance of PC1 lies between 40% and 50% and at around 20% for PC2 for all sets of correlations. The higher PCs explain lower shares of variance, less than 10% already for PC3 and only 1% for PC12.

Figure 10 provides the maps of the scores of first two principal components for the correlation between WS300 and precipitation and their monthly loadings. PC1 shows a pattern which looks like we have layered the winter pattern of the correlations in the western parts of the study region, the spring pattern of the northern region and the summer pattern of the TP. This is corroborated by the annual cycle of loadings for PC1. The highest positive loadings occur in winter and spring, which means that these months have the largest share on the dominant pattern. The loadings are high all year-round but highest in spring and lowest in July. This implies that this pattern is not very much influenced by the monsoon system or if so just in the onset period. We rather assume that the still relatively high loading is a result of the fact that we have high solar forcing and therefore convective activity on the TP which has a negative correlation to high wind speeds at 300 hPa. The loadings of the second PC (PC2) show a completely different annual cycle. They are highly negative in winter and highly positive in summer, just in the transition seasons the loadings are around zero. For this pattern winter and summer play a similar role, even with opposing arithmetic signs. This is an annual cycle pattern. The first two PCs together account for ~ 60% of the total variance of the data.

The first PC (PC1) of the correlation between WS10 and precipitation (Fig. 11a) shows a pattern which is dominated by winter and early spring situation, high positive scores in the PKwH region and the region around the Brahmaputra Channel and negative scores on the central and eastern TP. The loadings (Fig. 11b) are very high (~ 0.8) from November to April. Only in July and August the loadings are slightly negative. The positive correlations during summer on the





TP are not visible in PC1, they occur in PC2 which loadings have a direct opposing annual cycle, meaning high positive values in Summer and slightly negative in winter. The first two PCs together explain ~ 65% of the variance in the data.

All months exhibit high loadings for PC1 of the correlation between the vertical wind speed (both levels) and precipitation. Fig. 12 shows the scores and the loadings of the first two PCs for the correlation of W300 with precipitation, the PCs for the correlation between W10 and precipitation are not shown because of high similarity to W300. Summer and early autumn conditions have the largest impact, while loadings are lowest in winter, but still positive. Therefore, the high positive correlations in summer on the TP and in the lowlands south of the Himalayas are the dominant pattern, meaning that the vertical wind speed as a precipitation control is strongest effective during that time of the year, but has a mostly positive impact on precipitation year-round. PC2 exhibits much lower values. The annual cycle of the loadings shows, that winter and summer have both high loadings (~ 0.5), but with different algebraic signs, positive in summer and negative in winter. Spring and autumn exhibit very low factor loadings and are more or less only transition periods between winter and summer. The first two PCs together explain ~ 70% of the variance in the data.

The loadings of the first two PCs of the correlation between AWT and precipitation (Fig. 13b) exhibit a similar annual cycle like the loadings of the correlation between WS300 and precipitation. For PC1 the loadings are high (between 0.6 and 0.9) in winter, spring and autumn and low in summer (~ 0.2). This again emphasizes the finding that AWT is a precipitation control mainly in regions and seasons where moisture advection and frontal/cyclonic precipitation is dominant. The pattern of PC2 (Fig. 13a) is composed mainly by summer months (loadings > 0.8 in July and August), while the winter months exhibit negative loadings.

Regarding the correlation between PBLH and precipitation the dominant pattern PC1 (Fig. 14), which explains ~47% of the variance in the data, is dominated by the spring, which is clearly visible by comparing the pattern of PC 1 with the correlations in May (Fig. 8). All other months exhibit high positive loadings (>0.5), too, but August has the lowest share. The dominant pattern represents the spring pattern, which shows the transition between the winter (positive correlations) and the summer conditions (negative correlations). PC2 is mainly influenced by the high negative correlations dominant summer. The winter months exhibit high negative loadings. The different behaviour above the large lakes on the central TP is visible in both PCs.

In summary, the annual cycle of the loadings for each of the first two PCs, show a similar annual cycle. PC1 is mostly dominated by all seasons except summer, and mainly by the winter and spring situations. Whereas PC2 is determined by summer conditions. Winter conditions have also high loadings but with negative arithmetic sign, while spring and autumn only represent transition periods between these situations.

## 4 Discussion

### 4.1. General discussion of results

One main result of the current study are the high negative correlations between the horizontal wind speed in 300hPa and precipitation on the TP. This confirms the findings of Mölg et al. (2013), who showed that the flow strength at the 300 hPa level above the TP during the onset period of the Indian Summer Monsoon exhibits strong negative correlations with precipitation during this period in the regions influenced by monsoon, and explains 73% of the inter-annual mass balance variability of Zhadang glacier, located at the Nyainqenthangla range in the central TP. They only explain the correlations the other way round and argue that weak flow favours




convective cell growth. This together with the high positive correlations, in regions and seasons where frontal/cyclonic precipitation is dominant, detected in the current study shows the strong influence of the mid-latitude westerlies not only on the western parts of the study region but also on the TP itself, which was so far mostly described as mainly influenced by the monsoon

systems (Hren et al., 2009; Tian et al., 2007; Yang et al., 2014).
Previous studies stated that precipitation on the TP is controlled by the westerlies in winter and the Indian and East Asian summer monsoon in summer (e.g. Hren et al., 2009; Tian et al., 2007; Yang et al., 2014). This assumption is based only on precipitation timing, but Curio et al. (2015) and Mölg et al. (2013) already showed that the mid-latitude westerlies have also an impact on

summer precipitation. The current study highlights their findings by the detection of the negative influence of high horizontal wind speeds on precipitation development on the TP in summer.
The shown positive correlations of the wind speed in the boundary layer with precipitation in summer on the TP agree with the finding of Back & Bretherton (2005) who detected positive

correlations between near surface wind speed and precipitation only when convection can be triggered easily. Their study region was the Pacific ITCZ, but we assume that their findings are also valid for the TP in summer, where the convective activity is high and enough moisture is available at the surface.
The fact that the posititve correlation of AWT in summer on the central TP are not as high as

in the western parts of the study region in winter, means that here the convection is able to produce precipitation already with the moisture from local sources, which emphasizes the great importance of moisture recycling, like shown by Curio et al. (2015) using HAR data and by Chen et al. (2012), Joswiak et al. (2013) and Kurita and Yamada (2008), among others. Nevertheless, AWT has still a positive effect on precipitation, but it is not an essential control

during this time of the year on the central TP.
Spiess et al. (2015) showed that in summer increased horizontal wind speeds at 400 hPa have a positive effect on the height of the equilibrium line altitude at glaciers in different regions on the TP, they assumed that this could be caused by a reduction of convective precipitation due to high wind speeds. Exactly this process is shown by the strong negative correlations between

horizontal wind speed at 300 hPa and precipitation in summer on the TP (Fig. 3).

4.2 Sources of uncertainties

As always the uncertainty of the results mainly depends on the accuracy of the data itself, the

the aggregation of hourly data to daily means, and the methods used to analyse the data. The HAR precipitation and other variables, e.g. wind speed and direction, and temperature, were validated against other gridded data sets, global reanalyses and remote sensing data, and observations from weather stations by Maussion et al. (2011, 2014).
Due to the use of correlations we avoid problems regarding the exact precipitation rates falling

on the TP. Additionally the use of the monthly percentage on annual precipitation as input for the cluster analyses, to detect the precipitation seasonality in different regions of the TP, makes it possible to compare regions, which exhibit distinct different precipitation amounts. This is a general aspect to keep in mind, the decision to use mean daily data therefore has advantages and disadvantages. A major advantage is the possibility to analyse the data process based from

a climatological view, which could not be done on the basis of monthly data. But it is clear that some information is lost by aggregation. For example, it is not clear whether the negative correlations between the vertical wind speed and precipitation in the PKwH region are caused by a plausible physical process as described in section 3.2.2 or by the aggregation from hourly to daily resolution, which could mask out vertical wind speed conditions favourable for

precipitation due to information loss.





In general, when interpreting correlations, one has to keep in mind that correlation not automatically can be interpreted as causality.

The number of precipitation days on which all analysis depend is variable regarding the analysed months and regions, but we assume that the condition of at least 13 precipitation days
5 per grid point, applied for of all days of a specific month during the study period 2001-2013, respectively, ensures a reasonable data basis. Grid points which do not match this criterion are excluded from further analysis.

Since we only look at the first two PCs only, it is possible that mechanisms controlling precipitation, which appear in higher PCs only, are not considered in this study. The current
10 study is limited on dominant patterns and thereby to the first two PCs, because they together explain more than 60% of the variance of the data. Patterns with lower explained variance and transient patterns will be subject of a subsequent study.

## 5 Conclusion

15 Gaining a better understanding of the dynamic precipitation controls and the underlying processes is important since precipitation is the key element of the hydrological cycle of the TP and surrounding high mountain ranges and has a great impact on the water availability in the densely populated downstream regions, e.g. India, Pakistan and south-east Asia, by governing river runoff directly by precipitation or with a time lag by snow melt.

20 This study shows that different factors influence precipitation in different regions of the TP and adjacent high mountain ranges and during different times of the year in different ways, and sometimes even have an exactly opposite effect. For example, the wind speed at the 300 hPa level has a positive effect in the western parts of the study region in winter and spring, while it has a negative effect on precipitation on the TP during summer. This clearly shows that the
25 impact of the mid-latitude westerlies is strong, not only in winter by enhancing moisture advection for precipitation in the western parts of the study region, but also by cutting of deep convection during summer on the TP and in other regions and seasons where and when precipitation is mainly convective.

Therefore, the TP and the entire study region can be divided regarding the dominant form of
30 precipitation, cyclonic/frontal or convective precipitation. This enables us to more clearly realise the relevancy/importance of the monsoon system and the mid-latitude westerlies for precipitation distribution. The precipitation classification emerged by cluster analysis shows a mostly monsoonal influenced class, a convective class, and a hybrid class in between. This illustrates that it is not possible to draw an exact border for the monsoon extent and that there
35 will always be a relatively broad border area/region between monsoonal influenced precipitation and solely convective dominated precipitation caused by the inter-annual variability of monsoon strength and other factors.

Maybe one can say that the precipitation on the central TP in summer is influenced by the monsoon system regarding moisture supply, nevertheless moisture recycling is important, but
40 that the mid-latitude westerlies act as a control whether precipitation development is possible or not due the strong negative effect of high horizontal wind speeds on the development of deep convection.

In a next step we will analyse combined effects of precipitation controls, since the current study again showed that the controls are not independent from each other. We intend to use a
45 combination of PCA of the detected dominant patterns and cluster analysis to detect control regimes, like Forsythe et al. (2015) did to obtain a climate classification for the Himalayan arc and its surrounding. These regimes maybe can help to explain regional features of glacier mass balance, like the so-called Karakoram anomaly (e.g. Hewitt, 2005), or observed lake level





changes (e.g. Liu et al., 2010; Zhang et al., 2011), which show a different behaviour compared to the surrounding regions.

## Author contribution

5 J. Curio and D. Scherer designed the study and discussed all results. J. Curio carried out the analyses and prepared the manuscript with contribution from D. Scherer.

## Acknowledgements

This work was supported by the German Federal Ministry of Education and Research (BMBF) 10 Programme "Central Asia – Monsoon Dynamics and Geo-Ecosystems" (CAME) within the WET project ("Variability and Trends in Water Balance Components of Benchmark Drainage Basins on the Tibetan Plateau") under the code 03G0804A and by the German Research Foundation (DFG) Priority Programme 1372, "Tibetan Plateau: Formation – Climate – Ecosystems" within the DynRG-TiP ("Dynamic Response of Glaciers on the Tibetan Plateau 15 to Climate Change") project under the codes SCHE 750/4-1, SCHE750/4-2, SCHE 750/4-3.

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





Figures

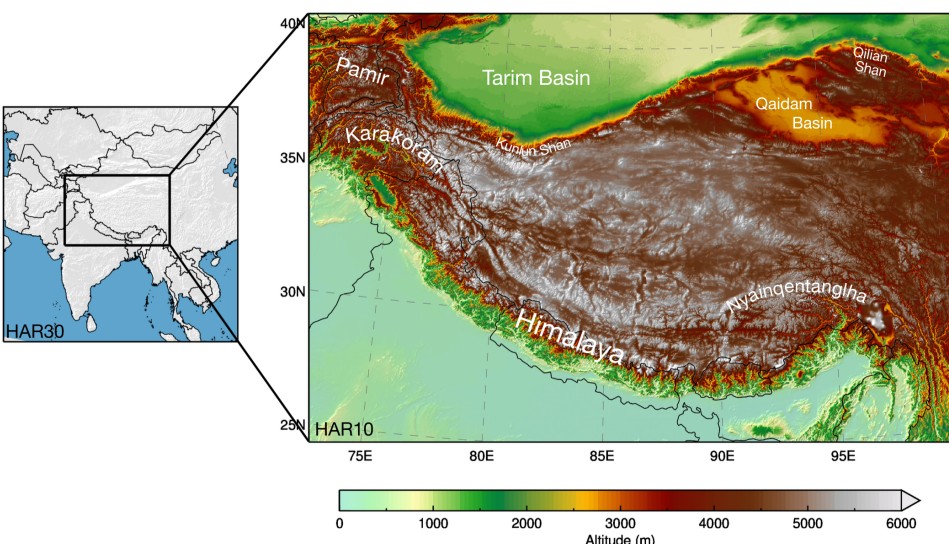

5    Fig. 1: Map of the Weather and Research Forecasting (WRF) model domain HAR10 (high Asia
      domain) and its location nested in the larger domain HAR30 (south-central Asia domain).
      Geographical locations are indicated in white.




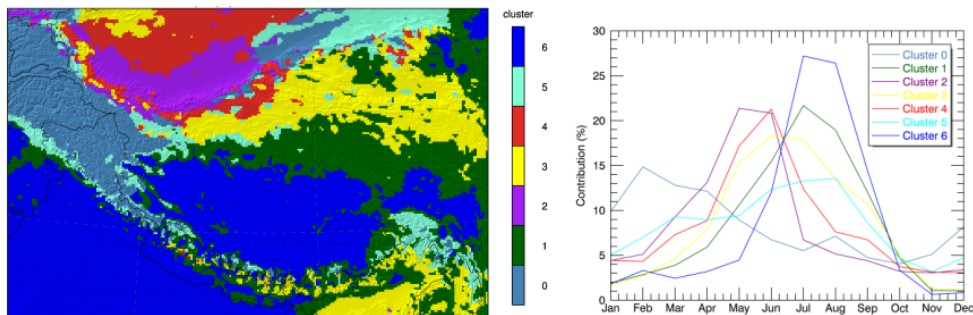

Fig. 2: Precipitation cluster (left) and the mean annual cycle of percentage contribution of monthly precipitation to annual precipitation for each cluster (right).

35

40

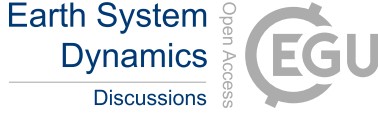



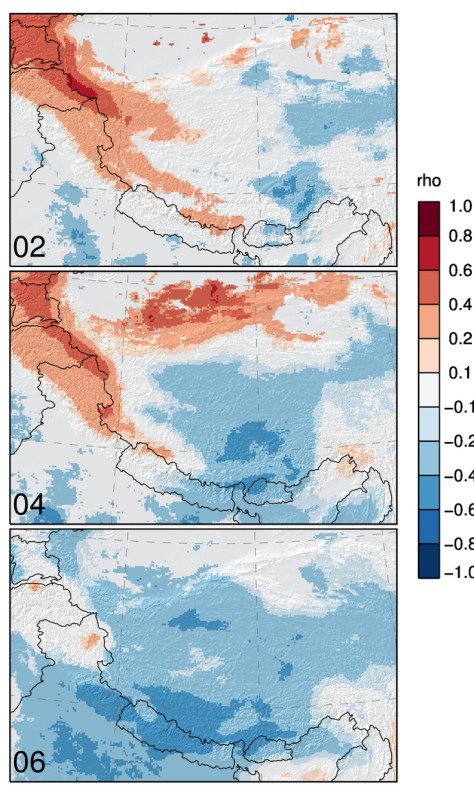

Fig. 3: Coefficient of correlation (rho) between horizontal wind speed at 300 hPa (WS300) and precipitation for February (02), April (04), and June (06). Positive correlations are denoted in red, while negative correlations are denoted in blue.





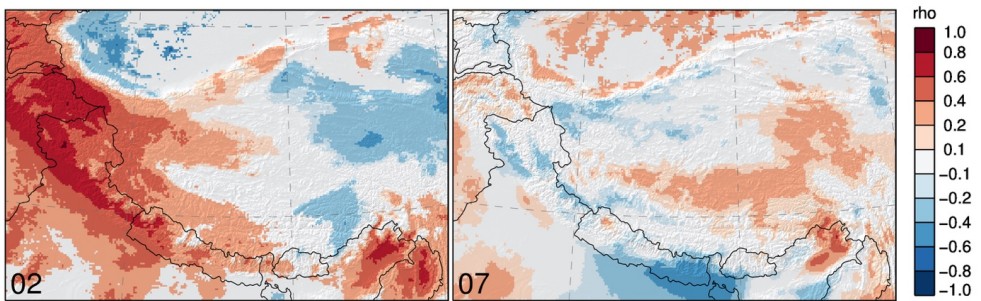

Fig. 4: Coefficient of correlation (rho) between horizontal wind speed at model level 10 (WS10) and precipitation for February (02) and July (07). Positive correlations are denoted in red, while negative correlations are denoted in blue.

35

40





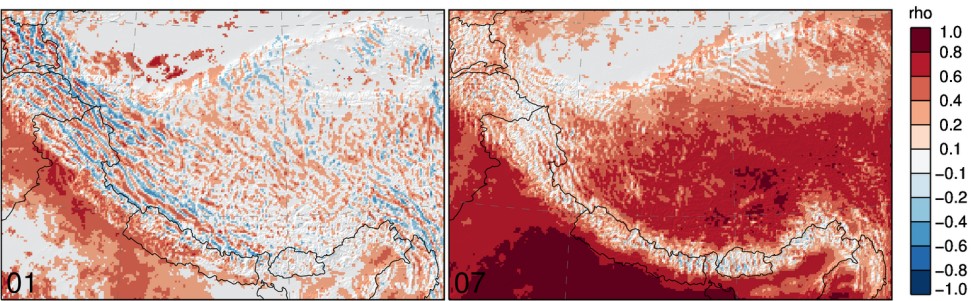

Fig. 5: Coefficient of correlation (rho) between vertical wind speed at 300 hPa (W300) and precipitation for January (01) and July (07). Positive correlations are denoted in red, while negative correlations are denoted in blue.

35

40





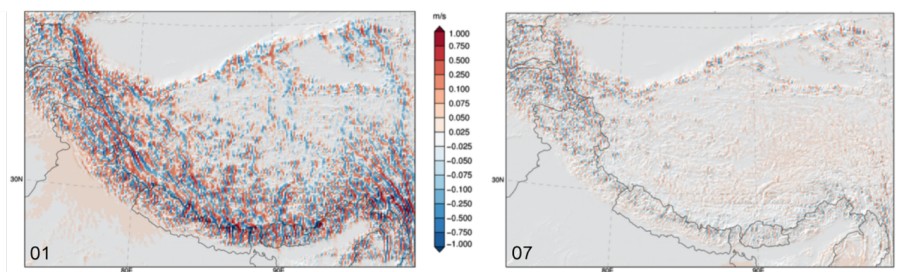

Fig. 6: Mean vertical wind speed at 300 hPa for January (01) and July (07).

35

40

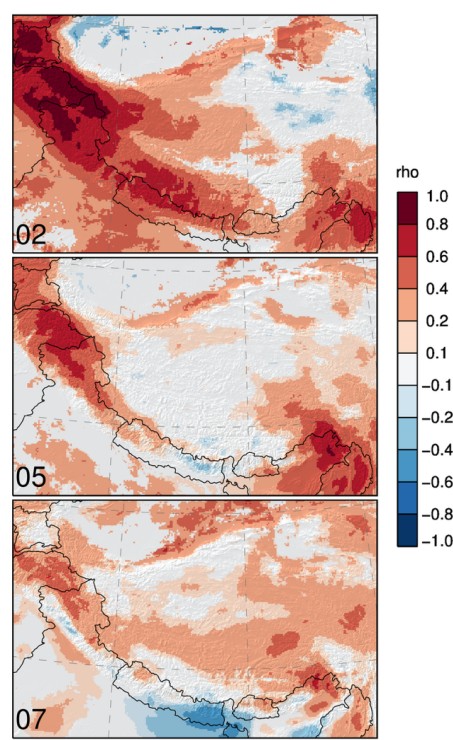

Fig. 7: Coefficient of correlation (rho) between column integrated atmospheric water transport (AWT) and precipitation for February (02), May (05), and July (07). Positive correlations are denoted in red, while negative correlations are denoted in blue.



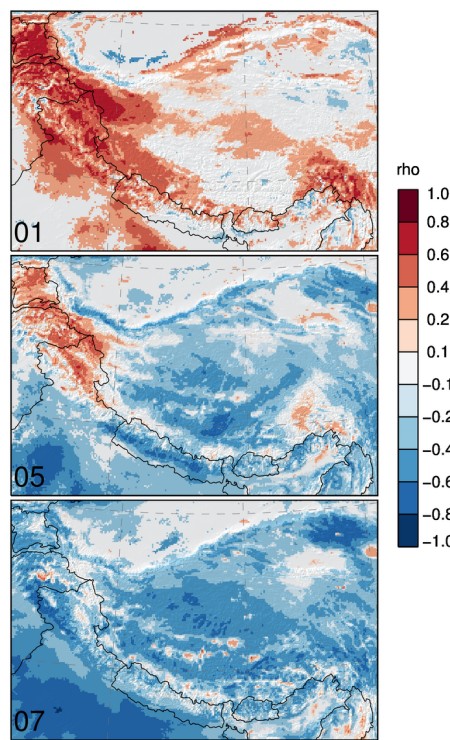

Fig. 8: Coefficient of correlation (rho) between planetary boundary layer height (PBLH) and precipitation for January (01), May (05), and July (07). Positive correlations are denoted in red, while negative correlations are denoted in blue.





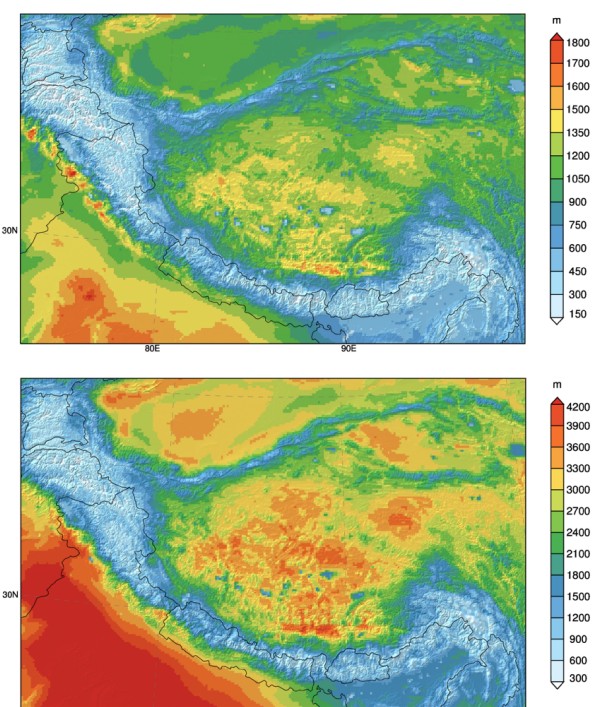

Fig. 9: Planetary boundary layer height (m) daily mean (top) and mean daily maximum (bottom) in May (2001-2013).



(a)

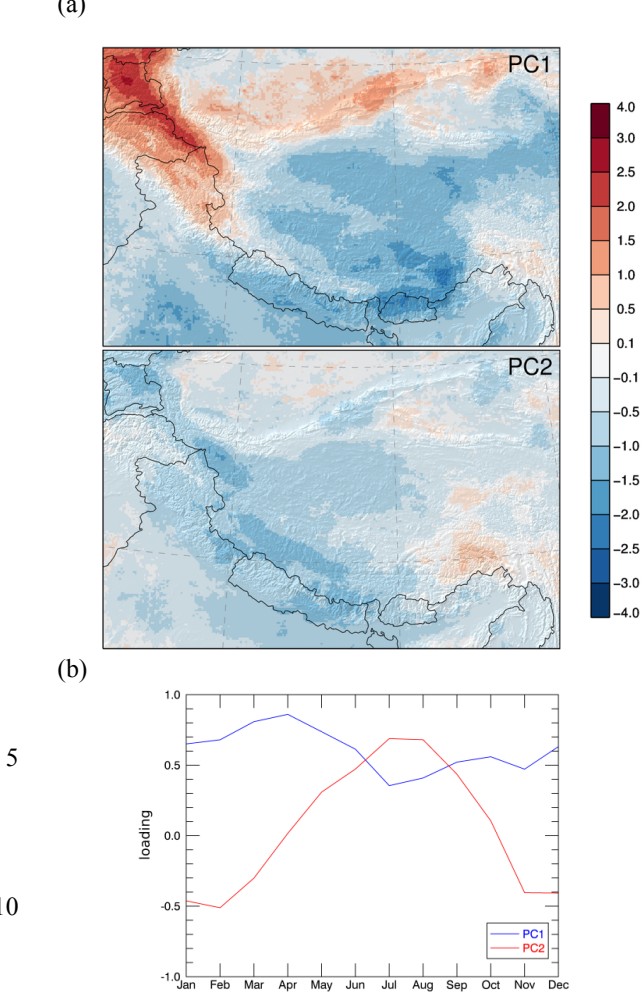

(b)

Figure 10. Scores of first two principal components PC1 and PC2 for the correlation between horizontal wind speed at 300 hPa (WS300) and precipitation **(a)** (positive values are denoted in red, while negative values are denoted in blue) and the monthly loadings **(b)** for both PCs.





(a)

(b)


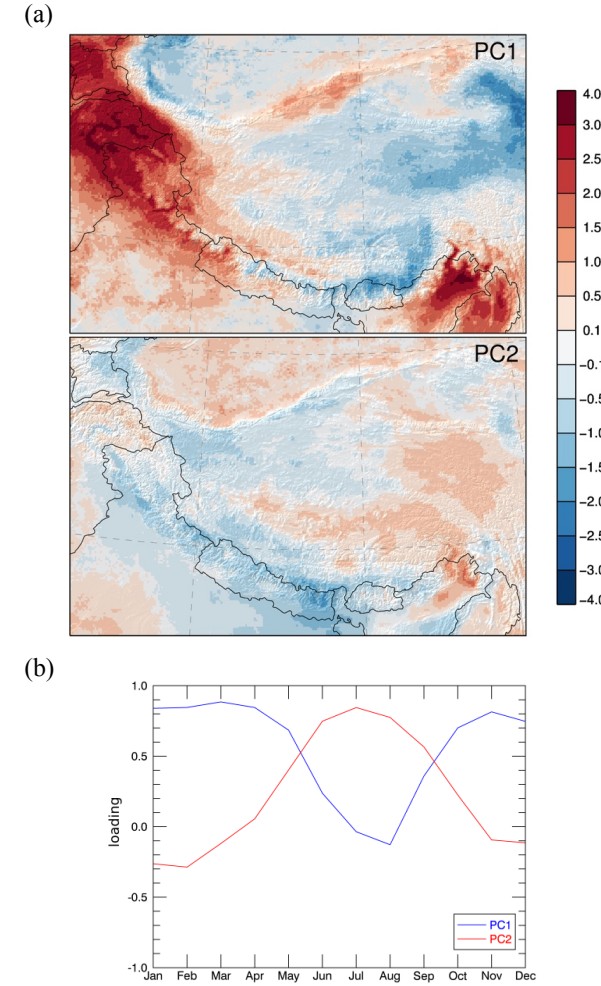

Figure 11. Scores of first two principal components PC1 and PC2 for the correlation between horizontal wind speed at model level 10 (WS10) and precipitation **(a)** (positive values are denoted in red, while negative values are denoted in blue) and the monthly loadings **(b)** for both PCs.






(a)

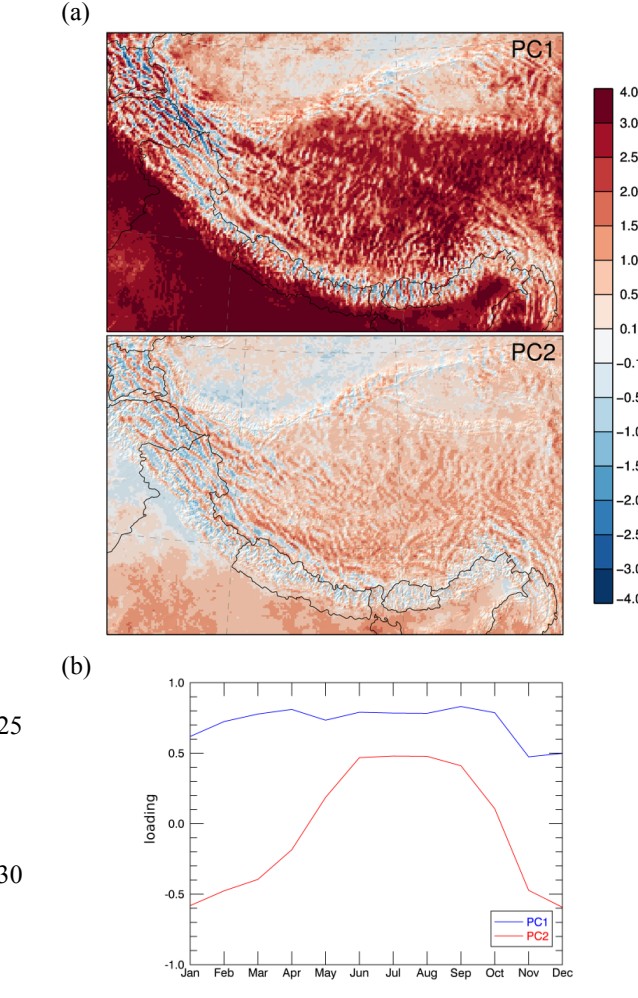

(b)


Figure 12. Scores of first two principal components PC1 and PC2 for the correlation between
vertical wind speed at 300 hPa (W300) and precipitation **(a)** (positive values are denoted in red,
while negative values are denoted in blue) and the monthly loadings **(b)** for both PCs.








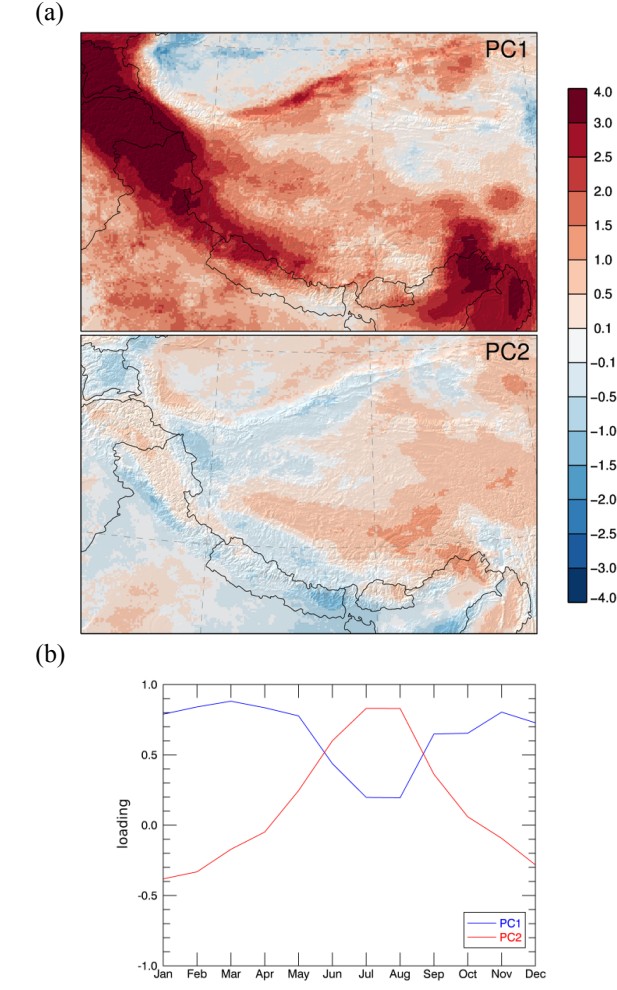

Figure 13. Scores of first two principal components PC1 and PC2 for the correlation between atmospheric water transport (AWT) and precipitation **(a)** (positive values are denoted in red, while negative values are denoted in blue) and the monthly loadings **(b)** for both PCs.



(a)

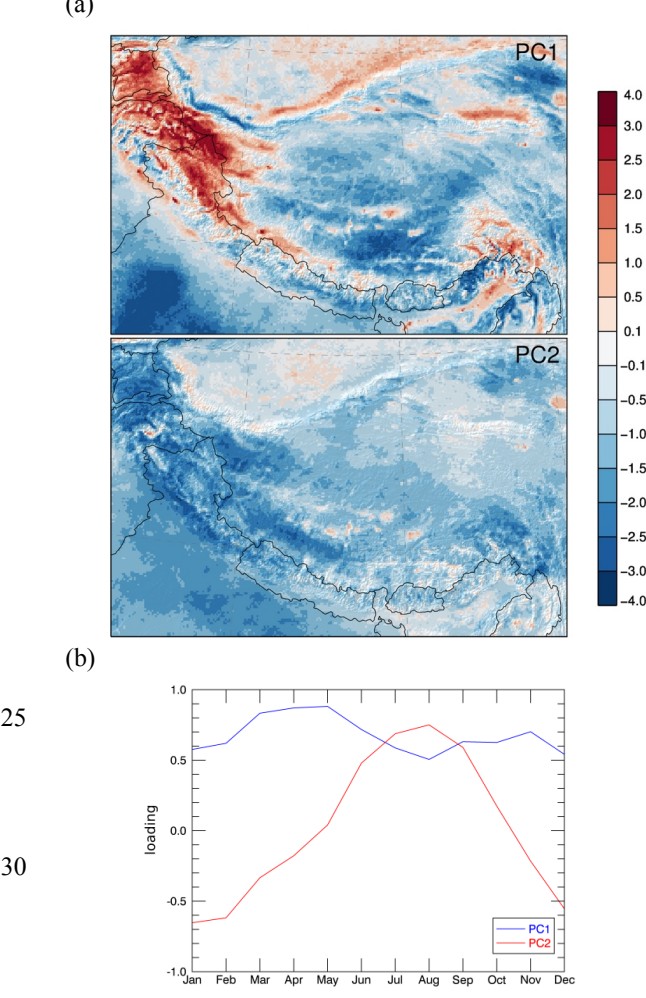

(b)

35

Figure 14. Scores of first two principal components PC1 and PC2 for the correlation between planetary boundary layer height (PBLH) and precipitation **(a)** (positive values are denoted in red, while negative values are denoted in blue) and the monthly loadings **(b)** for both PCs.

40