# Peer review of "Seasonality and spatial variability of dynamic precipitation controls on the Tibetan Plateau"

_Earth System Dynamics, 2016_

## Referee Comment (RC1) · Anonymous Referee #1 · 8 Mar 2016

Overview –

This manuscript investigates the influence of dynamic factors on precipitation in High Asia. Findings on the seasonality and spatial variability of precipitation controls (and precipitation type) in the region using horizontal and vertical wind speed, atmospheric water transport, and boundary layer height are an important contribution to the field. The research is presented in an intuitive manner and uses unique data to answer major questions that have not been previously addressed. I do have a few questions/concerns about the applicability of HAR, and am suggesting considerable revisions, but after these issues are addressed I believe the manuscript will be fit for publication.

General Comments –

[Figure]

HAR is a great data source that can help us address many questions that could not previously be asked in High Asia, but there are still many issues in modeling precipitation (especially convective precipitation) using 10km resolution with a convective parameterization. Rather than discourage the use of these unique data, I believe it is important for the current manuscript to address the uncertainty in HAR in a more substantial way. Referencing the papers is not sufficient in my opinion. Given that so much of the present analyses are based on correlations with convective precipitation in regions that have limited validation, it is important to thoroughly explain what the caveats are and why the precipitation controls describe here are robust despite uncertainty in the data.

Moisture variability is not sufficiently investigated in this manuscript. I would like to see atmospheric water vapor treated separately from advection where possible. Because the TP is so high, winds are generally very strong and moisture is minimal. Thus, the water vapor transport variable is much more heavily influenced by wind speed than water vapor. AWT figures in the manuscript confirm this bias by exhibiting heavy influences of the wind speed terms, which seem to mask any water vapor signal (even though the aren't independent). Furthermore, the opposite seasonal cycles of wind and moisture, and their respective sources of variability, may be illustrative of changes in precipitation controls through the year.

The precipitation/pblh relationship over the TP is something that I do not understand well, and would like to see more about, however, this discussion elicits questions about the PBL parameterization that was used in HAR, and it's validity over the TP that can't be answered in this paper. Though Maussion et al. (2011) performed a few PBL sensitivity tests, validation was not done to specifically evaluate PBLH and thus it is impossible to determine how representative this variable is of the actual conditions. Though this is somewhat true for the other precipitation controls used in this study, I am especially skeptical of PBLH because it is not resolved explicitly, and is so poorly represented in other parts of the world. In my opinion, the discussion is an interesting
but non-essential part of the paper, and ultimately, I think it is best to remove it.

There are a number of careless grammatical errors and poorly structured sentences in this manuscript that distract the reader. I've noted a few of instances of sentences that were difficult to read in the "technical comments" section, but I encourage the authors to carefully edit the full text.

Specific Comments –

Page 2, Line 10: Why was vertically integrated water vapor transport analyzed and not column integrated water vapor? Since you are already investigating winds, it makes sense to investigate moisture without accounting for wind so that you can determine whether moisture availability is due to advection or local changes in temperature/evaporation? The two aren't completely independent, but they do have opposite seasonal cycles as well as their own sources of variability.

Page 2, Line 49-51: Referencing a study of deep tropical convection over the ocean hardly seems relevant to precipitation controls over the TP. It would be better to have a reference for mountain environments, or at least for land areas. If not, it may be necessary to show this relationship using HAR. Furthermore, this sentence is at odds with the negative effect of high horizontal wind speeds on convective precipitation that is described in following sentence.

Page 3, Line 4: Note that the real benefit occurs because of orographic forcing of the moist flow. Without topography the relationship would be different.

Page 3, Line 15: It may be interesting to look at atmospheric stability (beyond the PBLH). In convective precipitation, vertical motions occur for different reasons than during frontal precipitation. Looking only at stability (theta or theta-e) would be instructive as to environmental conditions that support vertical motion irrespective of horizontal wind against the orographic barrier. This is important for both summer and winter. (this is only a suggestion)

Page 3, Line 41: This raises a number of questions for the reader: Were multiple PBLH schemes tested for WRF? What confidence is there that the PBLH from the configuration with the chosen parameterization is performing well? Were PBLH scheme sensitivity experiments performed in the development of HAR and if so, what observations are they validated with? If PBLH is to be included in this paper (which I've already argued against in the general comments section) it is necessary to address these very important questions in the manuscript rather than only providing the reference for HAR.

Page 4, Line 10: Because this study focuses on convective precipitation, some discussion of the validity of the HAR dataset, which uses a convective parameterization, should be included here. Beyond the reference, the authors should clearly note that precipitation during summer is highly sensitive to the choice of convective parameterization. Furthermore they should note that using higher resolutions and resolving precipitation explicitly may alter the spatial and temporal distribution of precipitation. Despite the issues associated with HAR precipitation, I do believe that it is very useful for understanding the mechanisms presented in this paper and the spatial and temporal differences in precipitation across large regions of the TP. I am simply encouraging the authors to further discuss the caveats here so that the reader is aware of these issues. After all, they may not be familiar with the Maussion articles or may not read them to better understand the data that you have used.

Page 4, Line 32-33: I am curious as to how many days are included in the precipitation analysis. It seems that a threshold of 0.1 in the Karakoram and western Himalaya would remove very few days from consideration through the winter months. Why was a percentile-based threshold not used in addition to this low threshold to further exclude days with light precipitation that don't really contribute to seasonal totals in an area such as the KH. Including these dates masks the signal of precipitation controls during events that contribute strongly to overall precipitation accumulation (e.g. in the Karakoram and western Himalaya only a few dates during the winter season contribute the majority of seasonal (and in some cases, annual) precipitation).

Page 4 Line, 43-44: Do correlations in regions of sparse precipitation exhibit considerable sensitivity to a few extreme events? If you threshold out the largest few events (99th percentile) would the correlations change?

Page 5, Line 20-22: The yellow and green classes may be very sensitive to the cumulus parameterization scheme that was used in HAR. Due to the uncertainty in how well WRF, and specifically HAR in this case, represent convective precipitation in the Himalaya, I don't think these classes can be so clearly defined. At the very least, better discussion of the caveats must be given, but it may be better to not distinguish these groups since the precipitation controls discussion does not.

Page 6, Line 1: It seems to me that the reason the high elevations of the eastern Himalaya (near the Brahmaputra Channel) exhibit considerable precipitation contributions outside of the monsoon season is due to the orographic locking of westerly flow and the large amount of available moisture in this area. See Norris et al. (2015; their Figs. 5&6), which are focused on westerly disturbances affecting other regions of the Himalaya, but in both cases, exhibit precipitation in the eastern Himalaya generated by terrain locking of westerly flow.

Page 6, Line 30-32: This should cite some of the more seminal work on westerly disturbances.

Page 6, Line 32-33: Higher wind speed does not necessarily mean enhanced moisture supply. There are many factors that modulate both wind and moisture in these systems (see Cannon et al. 2015). I would suggest just removing "which benefits from enhanced moisture due to higher wind speeds" and stating that the cyclonic/frontal precipitation is associated with westerly disturbances and then add a citation (e.g. Dimri et al. 2015 – review paper of westerly disturbances).

Page 7, Line 2: Figure 1 indicates that the western Himalaya receives more than 0-5% of annual precipitation during summer (June alone is over 5% for Cluster 0). Furthermore, some studies argue that about 30%-60% of precipitation in this region falls

during summer (e.g. Bookhagen and Burbank, 2010; their Fig. 4). Though the Bookhagen and Burbank estimate is probably too high because it is based on TRMM, which doesn't do well with winter precipitation, it is clear that considerable rainfall occurs in the western Himalaya during summer. Given that there is precipitation, what then could explain the lack of correlation?

Page 7, Line 7-8: Remove the linkage between higher wind speed and enhanced moisture. I would argue that higher wind speed more efficiently extracts moisture due to stronger orographic forcing. As mentioned before, there are many controls on moisture availability in the western Himalaya that are independent of the cross-barrier wind speed.

Page 7, Line 32-39: The alternating positive/negative correlations look more like a gravity wave (i.e. updrafts and heavy precipitation on windward side of the mountain, downdrafts and light precipitation in the lee; Roe et al. 2005) than an error associated with aggregation. There should be an easy way to show this using topography aspect and wind direction, or using higher temporal resolution data available from HAR.

Page 7, Line 41: I recommend looking at the correlation of precipitation and precipitable water rather than precipitation and water transport (or at least in addition to it). This is particularly important to do since convective precipitation requires enhanced moisture content, but not enhanced wind, while orographic precipitation requires both enhanced wind and enhanced moisture. Even though wind and moisture aren't independent, it would be interesting to see a figure of just the precipitable water correlations since it does not include the advection term, for which a similar variable was shown in previous figures (WS300, WS10). This could be particularly illustrative since the seasonal cycles of moisture availability and wind speed are opposite for this region (e.g. Cannon et al. 2015).

Page 8, Line 31: I don't see the need to introduce boundary layer height, which is controlled by other processes that have already been evaluated. I don't think this adds

much to the discussion, which was already quite strong. Furthermore, PBLH is parameterized in WRF, so you're introducing another question about how well this parameterization works, rather than focusing on variables that are explicitly resolved (moisture, temperature, wind). Though Maussion et al. 2011 performed a few sensitivity tests with different PBL parameterizations, validation specific to PBLH using radiosondes or wind profilers over the TP has never been done, so there are a lot of unknowns here. The lakes discussion is certainly interesting, but given the uncertainties about WRF's ability to resolve these processes, and because this discussion is not a main topic in the paper, I suggest leaving it out.

Page 11, Line 33: The uncertainties in HAR should include the use of a convective parameterization, which directly affects the precipitation data (presence, timing and magnitude) that this study is based on. A better description of this particular issue is required since it is possible that some of the results in this work are sensitive to the choice of parameterization (or that if HAR were performed to explicitly resolve convection (<6km)).

Page 11, Line 48: Did you try using hourly or 3-hourly data to test what role aggregation errors play in your research? It seems that you have the necessary data to directly address this uncertainty.

Technical Comments –

Page 1, Line 7-8: Change "moisture" to "water resources"

Page 1, Line 32: Change "strengthen" to "strengthening"

Page 2, Line 1-8: A few of these sentences were difficult to read. A little restructuring would help the reader here.

Page 2, Line 25: When introducing the acronyms (PBLH, AWT and WS300), they should first appear next to their full names.

Page 3, Line 7 & 18: "luv side" should be changed to "windward" as that is the common

terminology in meteorology textbooks. (change in all instances)

Page 4, Line 24-25: Did you aggregate daily, or are you using a specific time-slice for each day?

Page 5, Line 21: Change "intensive" to "pronounced". Intensive makes it sound as though this relates to magnitude of precipitation values rather than the shape of the distribution.

Page 6, Line 13-14: remove "and therefore exhibit no coherent patterns". The patterns are coherent when considering mesoscale, synoptic and large-scale influences as well as topography. The patterns are complex and heterogeneous, but not incoherent.

Figure 2: The cyan and yellow lines are difficult to see. Perhaps all the lines could be thicker or darker colors could be used

Page 7, Line 12: "There are high positive correlations (over the Tibetan Plateau?) between WS10 and precipitation. . ."

There are too many small mistakes in word choice, grammar and sentence structure to identify each individually. I encourage the authors to carefully edit the full text.

---

## Referee Comment (RC2) · Anonymous Referee #2 · 21 Mar 2016

This manuscript analyzed the seasonality and spatial variability of dynamic precipitation controls on the Tibetan Plateau using HAR dataset. It stresses the high impact of the mid-latitude westerlies on precipitation distribution on the TP and its surrounding year-round. I can feel the strong eagerness of authors on concluding the westerly is the controller of the precipitation over the TP. However, manuscript is supportive enough. I have great concerns before the conclusions could be drawn. Substantial revisions are necessary before the manuscript is publishable.

All of this work is based on the single approach – correlation and single data set – HAR. Large uncertainties in conclusions occur due to the approach adapted and data set used. Multiple approaches or datasets are necessary to swipe away these uncertainties.

[Figure]

More validation and evaluation on HAR are of fundamental necessity before it could be used on analyzing. The whole basement of this study is the precipitation classification by Maussion et al. (2014). However, the precipitation classification Maussion et al (2014) did is for only the glacier accumulation regimes locating at high altitudes above about 5000m shown in their Fig. 14, rather than for the whole Tibetan. Is it representative for the precipitation over the whole TP? If so, please show the evidences. If not, suggest changing the title to "…on the glacier accumulation regimes over the Tibetan Plateau"

P2L47, it reads "on average, more than 60% of moisture needed for precipitation falling on the inner TP are provided by the TP itself (Curio et al. 2015)". In authors' previous paper published in 2015. That suggests that convections over the TP dominate precipitation in the TP rather than the moisture transportation from outside. In this manuscript, the westerly are argued to be the dominant controller in precipitation. These two conclusions are conflict to each other. Which one is the leading controller of the precipitation in the TP, in authors' ultimate view? What is the linkage of the convections and the westerly? Considerate analysis and evaluation are strongly suggested before the conclusion is drawn.

Over ocean or area with low elevations, 300 hPa is high enough to stand for the height that the westerly locates. However, the TP possesses an elevation above 4000m on average. 300 hPa is too low for the westerly over there. Authors cite Schiemann et al. (2009) as the reason of this selection. We can read the cited reference is about the precipitation climate of Central Asia. The TP possesses distinguish climate from the Central Asia not only in its unique height, but also the distance from the ocean. They are not comparable. Authors should refer to works in the TP rather than other where. Numerous studies claim that the westerly reaches as high as 100 hPa over the TP. For instance, 200 hPa (in the global climate model domain, Gao et al., J. Climate 2014) or 100 hPa (in regional climate model domain, Gao et al., J. Climate 2015) are the height where the westerly hang over the Tibetan; whereas, the 600 hPa (in the

global climate model domain) or 500hPa (in regional climate model domain) is the near surface. The 300 hPa is a middle layer between the upper and near surface layers. It is reasonable using the vertical wind speed at 300 hPa for the vertical motions. However, the horizontal wind speed at 300 hPa used to represent the westerly jet over the Tibetan is questionable.

It is claimed that "Six controllers are selected". However, five are analyzed in 3.2.1a, 3.2.1b, 3.2.2, 3.2.3 and 3.2.4. Do I missing something? Before the controller is concluded, I prefer to call them elements rather than controller. In addition, background of these elements is missing. Why these elements are chosen? What is their relevance with precipitation? For instance, the horizontal wind speed at model level 10 (WS10) is used. What is the height of model level 10? What it stands for? Is the PBLH relevant to PBL parameterization schemes used in simulation?

---

## Referee Comment (RC3) · Anonymous Referee #3 · 21 Mar 2016

Summary:

The study described in this manuscript investigates the impact of five selected dynamic controls (horizontal wind speeds at different levels, vertical wind speed, atmospheric water transport and planetary boundary layer height) for precipitation in High Asia. The study's novelty lies in the use of a relatively new high resolution dataset to answer questions that have previously not been addressed. The conclusions reached are supported by the findings. They are scientifically valuable and presented in a logical and intuitive way. The title and abstract are concise. However, I do have a few questions, concerns and suggestions regarding the methods applied and language used in the manuscript. After consideration of those and moderate revisions I recommend publication of this manuscript.

[Figure]

General Comments:

The statistical methods used in this study are suitable for addressing the questions. However, since the findings and conclusions rely heavily on purely statistical methods (mostly Spearman's rank correlation), I would like to see them discussed in more detail (see specific comments). The authors should also describe how the significance of rho was calculated. (I believe this is missing entirely from the manuscript). The PCA is suitable for describing the spatio-temporal variability of correlation of controlling factors and precipitation.

I am not sufficiently familiar with specific shortcomings of the HAR dataset to comment on whether or not the authors adequately addressed these in the manuscript and took them into consideration when drawing conclusions. However, since this study is entirely based on HAR, I would appreciate some comments on existing uncertainties of the dataset and how/whether or not this limits the interpretation of the results of this study.

There are numerous grammatical errors throughout the manuscript and sentence structure is often confusing. This makes it very difficult to read and understand in certain sections. I highlighted some of these in the "technical comments" below. I strongly recommend the authors edit the language of the manuscript and let a native speaker (or someone with a similar level of written English) review it before resubmission.

Specific Comments:

Page 2, line 8: You write here and later on that you select six factors. However, you only list five here (and in the results). Maybe I am missing something?

Page 2, line 15: What is this assumption based on? I expected a little more explanation for the selection of controlling factors – scientific or purely technical. Also, what factors were excluded and why?

Page 4, line 33: How was the 0.1mm threshold chosen? In context of the study's aims, what are the advantages of choosing an absolute value instead of a grid-box specific

percentile for example?

Page 4, line 37: It may be more insightful to highlight the advantages of the Spearman rank correlation over other measures of statistical dependence for this particular question involving precipitation and its controlling factors.

Page 4, line 39: There are multiple ways in which the statistical significance of such a correlation can be determined. I recommend that you at least mention in one or two concise sentences how it was determined in this study. Also, how sensitive are your results to different, commonly used significance levels, e.g. 0.01? Since the correlation analyses form the centre piece of your study (and the PCA's are also based on correlation coefficients), I think it is necessary to provide a little more insight.

Page 5, line 1-3: Maussion et al. use this clustering approach to classify glacier accumulation regimes. While the analysis itself is a universal tool of of descriptive statistics fitting for the aims of the study, I recommend the addition of justification for deviation from the Maussion et al. clustering since you present this as something that builds on that study. For example, why choose seven instead of five clusters as Maussion et al. do? (see comment below)

Page 5, line 5-8: As I understand, you varied k for the clustering to determined the optimal number, i.e. the number giving you good coherence within the classes and sufficient distinctions between them. What was the k range you used and how did you determine optimal k? Was this apparent from a qualitative assessment of plotted results or was it determined by something like a discriminant analysis? Furthermore, for similar clusters such as purple/red and yellow/green, I recommend adding comments on the sensitivity of the clustering to physical conditions versus HAR/WRF limitations. In other words: is the separation of yellow and green physically meaningful?

Page 7, line 32: Could this pattern be the result of a gravity wave?

Page 12, line 1-2: This is a very general comment and does not go into the type of

correlation you are dealing with in this study. I believe Spearman's R to be adequate here, but I recommend adding a word of caution, since a nonparametric measure for correlation limits the interpretation of R values. It is not a source of uncertainty as such, but should be mentioned somewhere in the manuscript (along with more details on the significance tests and the sensitivity of results to significance levels)

Technical Comments:

There are too many grammatical errors to highlight all in this section. Furthermore, the phrasing of many sentences is confusing. I strongly suggest to let a native speaker to review the language of this manuscript.

Page 1, line 33: "strenghtening" instead of "strengthen"

Page 1, line 31-33: This sentence is confusing. Think about rephrasing it or breaking it up into two sentences.

Page 2, line 3: Do you mean "gives us the opportunity for a process based analysis of the data"?

Page 2, line 5-7: Change to something like "Therefore, we want to examine the timing, location and strength of the influence of precipitation controls on precipitation development."

Page 2, line 7-8: Rephrase to something like "The aim of this study is to describe the spatial and temporal correlation of [...]"

Page 2, line 12: I suggest citations at this point when making statements about the influence of the factors being known.

Page 2, line 25: Abbreviations should be introduced earlier in the text when their full names are first mentioned.

Page 2, line 33: Change to "correspond to"

Page 2, line 35: Change to "at a height"

Page 2, line 36: "[. . .] which strength and location [. . .]" - review the grammar here/rephrase.

Page 3, line 7: "windward side" is more commonly used in English than "luv side" (which is still common in German literature). I recommend changing it throughout the manuscript.

Page 3, line 45-47: This is not gramatically sound (see comment Page 2, line 12)

Page 4, line 1-2: This is not gramatically correct.

Page 5, line 19: Correct to "[. . .] precipitation is falling in this region, [...]"

Page 5, line 25: "There" should be the start of a new sentence for this to be grammatically sound.

Page 5, line 30-33: This sentence is confusing and not grammatically correct. Please rephrase.

Page 5, line 35: Correct to "its"

Page 5, line 44: Correct to "evenly" (adverb)

Page 7, line 30: Do you mean "This supports the interpretation [...]"?

Page 8, line 4: Correct to "Therefore"
* * *

---

## Referee Comment (RC4) · Anonymous Referee #4 · 23 Mar 2016

This paper by Julia Curio and Dieter Scherer try to demonstrate that the westerlies is an important factor influence the precipitation in the Tibetan Plateau even in the monsoon season, based on the High Asia Refined analysis. The correlation and principal components are used in this manuscript. There are five major problems in the manuscript described below. By considering these issues, I suggest major revisions.

Major issues: 1. The text is not fairly organized, especially the introduction. The discussion and interpretations are superficial given the figures results. Authors need to present their work with better clarity.

2. Authors used the HAR as the unique data for analyses, but they did not evaluate the dataset with observations on the Tibetan Plateau, and did not confirm their results with other data. The systematic analysis of stable isotopes in precipitation on the Tibetan

[Figure]

Plateau has demonstrated the seasonal moisture origins and moisture transports. I suggest the authors to refer it.

3. Why these six factors are considered in this study? Winds does not mean the precipitation if there is no moisture transport with them. Authors concluded that all of these factors combined influence on precipitation. In this case, what is the main contribution of this study? Authors also said moisture recycling was important that offers more than 60% moisture for precipitation. Which one is more important, the westerlies or the recycling? It is unclear.

4. Why is 300 hPa for the westerlies? Authors did not clarify the precipitation heights on the Tibetan Plateau. Due to the complex topography and climate, different type of precipitation shows diversified contribution to the annual precipitation amount and it occurs at different height. It should be considered.

5. This manuscript is not easy to follow because of many grammatical errors and disordered sentences.

---

## Author Comment (AC1) · 3 Jun 2016

**Author response**

**General response**

First, we would like to thank the four anonymous referees for their helpful and thoughtful comments, remarks and suggestions. We are sure that these will help to improve the manuscript. We will add a description of validation and evaluation conducted for HAR to a revised version of the manuscript. We will carefully revise the whole manuscript to improve its structure and language. We revised some of the sentences marked as being grammatically incorrect by the reviewers. Since we have to rephrase parts of the manuscript during the process of revision, we cannot promise that we will not have to change them again, and want to make the reviewer and editor aware of this possibility. The reviewer comments are highlighted in black, while our responses are highlighted in blue. We numbered the reviewer comments to make it easier to refer to a response to a specific comment later in the author response. R1C1, for example, stands for reviewer 1, comment 1.

Anonymous Referee #1

Overview –

This manuscript investigates the influence of dynamic factors on precipitation in High Asia. Findings on the seasonality and spatial variability of precipitation controls (and precipitation type) in the region using horizontal and vertical wind speed, atmospheric water transport, and boundary layer height are an important contribution to the field. The research is presented in an intuitive manner and uses unique data to answer major questions that have not been previously addressed. I do have a few questions/concerns about the applicability of HAR, and am suggesting considerable revisions, but after these issues are addressed I believe the manuscript will be fit for publication.

General Comments –

R1C1: HAR is a great data source that can help us address many questions that could not previously be asked in High Asia, but there are still many issues in modeling precipitation (especially convective precipitation) using 10km resolution with a convective parameterization. Rather than discourage the use of these unique data, I believe it is important for the current manuscript to address the uncertainty in HAR in a more substantial way. Referencing the papers is not sufficient in my opinion. Given that so much of the present analyses are based on correlations with convective precipitation in regions that have limited validation, it is important to thoroughly explain what the caveats are and why the precipitation controls describe here are robust despite uncertainty in the data.

AR: Validation and evaluation of HAR: The HAR data set was validated by Maussion et al. (2014) by comparison with rain-gauge observations from the National Climatic Data Center (NCDC) and the satellite derived gridded precipitation data from the Tropical Rainfall Measuring Mission (TRMM). The HAR shows a slightly positive bias in comparison with station data, 0.17 mm/day for HAR10 (0.26 mm/day for TRMM

3B43 product). The comparison with TRMM shows that HAR captures the general features of precipitation seasonality, variability, and spatial distribution. Maussion et al. (2014) found that the HAR10 precipitation averaged over the domain shows 15% more precipitation than TRMM, but these differences are assumed to be related to the well-known underestimation of snowfall and light rain by TRMM. Convective precipitation is simulated in agreement with results from literature, but one has to keep in mind that the model uses a parameterization scheme for cumulus convection. A spatial resolution of 10 km is not high enough to resolve cumulus convection. The HAR is able to reproduce orographic precipitation features as documented by Bookhagen and Burbank (2010). Maussion et al. (2014) state that these qualitative considerations cannot provide a quantitative uncertainty value. Curio et al. (2015) compared the HAR30 atmospheric water transport (AWT) with ERA-Interim. They found similar patterns; the differences being related to the different spatial resolutions and thus a better representation of the underlying topography by the HAR. In the HAR data, the blocking of AWT from the Bay of Bengal by the Himalayas is more pronounced, and the results show the importance of meridionally orientated high mountain valleys for moisture supply to the Tibetan Plateau. For a revised version of the manuscript, we have compared the 300 hPa wind of the HAR with ERA-Interim. Figure R1 shows that they are in a good agreement with each other. Due to the daily reinitialization strategy used to generate the HAR data set, the wind fields in higher levels cannot evolve far away from the forcing data as this is possible for longer model runs. The question how large the uncertainties of the HAR data and especially precipitation are and how we can estimate them, is a topic which should be investigated more in detail. Since there are no other gridded data sets with a comparable high temporal and spatial resolution, the possibilities to validate the HAR are generally limited and will be subject of future research.

[Figure]

Fig. R1: Mean wind speed at 300 hPa for January (left column) and July (right column) for HAR10 (top) and ERA-Interim (middle) and the differences between them (bottom).

Additionally, we plan to conduct another analysis during the revision of the manuscript to validate the precipitation clusters obtained from the HAR data against NCDC precipitation data. We want to detect whether the stations will occur in the same cluster as the nearest HAR grid points or will be associated to a different cluster regarding their precipitation timing and amount.

R1C2: Moisture variability is not sufficiently investigated in this manuscript. I would like to see atmospheric water vapor treated separately from advection where possible. Because the TP is so high, winds are generally very strong and moisture is minimal. Thus, the water vapor transport variable is much more heavily influenced by wind speed than water vapor. AWT figures in the manuscript confirm this bias by exhibiting heavy influences of the wind speed terms, which seem to mask any water vapor signal (even though they aren't independent). Furthermore, the opposite seasonal cycles of wind and moisture, and their respective sources of variability, may be illustrative of changes in precipitation controls through the year.

AR: We selected the atmospheric water transport and not the column integrated moisture content because this study focuses on dynamic precipitation controls. This means that we focus on dynamical processes that have an influence on precipitation variability and not on the causes of precipitation. Advection of moisture is one of these dynamical processes. The term control means that these variables have a modifying influence on the average precipitation conditions; they can either increase or decrease the precipitation amounts. This shows that there is a covariance between the controls and the precipitation and therefore strong correlations. These correlations are independent from the mean values of precipitation since we use Spearman rank correlation where the correlation is computed using the ranks of the values and not the actual values.

In summer we have high amounts of AWT and the wind speeds are much lower than in winter but we still have high positive correlations with precipitation. We do not think that the correlation figures in the manuscript are in general biased by the wind speed term.

Additionally we calculated the vertically integrated atmospheric moisture content for the HAR and repeated the calculation of correlation for this variable. The result (figure R2) shows high positive correlations between the atmospheric moisture content and precipitation throughout the year, as expected. We plan to add this figure and description to a revised version of the manuscript or to the supplement.

[Figure]

Fig. R2: Coefficient of correlation (rho) between the vertically integrated atmospheric moisture content and precipitation for all months (01-12). Positive correlations are denoted in red, while negative correlations are denoted in blue. Only correlations significant at the 0.05 significance level are shown.

R1C3: The precipitation/pblh relationship over the TP is something that I do not understand well, and would like to see more about, however, this discussion elicits questions about the PBL parameterization that was used in HAR, and it's validity over the TP that can't be answered in this paper. Though Maussion et al. (2011) performed a few PBL sensitivity tests, validation was not done to specifically evaluate PBLH and thus it is impossible to determine how representative this variable is of the actual conditions. Though this is somewhat true for the other precipitation controls used in this study, I am especially skeptical of PBLH because it is not resolved explicitly, and is so poorly represented in other parts of the world. In my opinion, the discussion is an interesting but non-essential part of the paper, and ultimately, I think it is best to remove it.

AR: Thank you for this suggestion which we will follow. In a revised version of the manuscript PBLH will no longer be included. We still find the PBLH a very interesting variable, because it gives information about the turbulent mixing of the lower atmosphere and convection. The PBLH itself is not a pure dynamic variable but entrainment plays a big role and the PBLH is determined dynamically in the HAR data. The PBLH is not the best choice to analyze precipitation controls since it is itself

controlled by other variables. Renouncing the analysis of the PBLH helps to keep the manuscript more focused and concise.

R1C4: There are a number of careless grammatical errors and poorly structured sentences in this manuscript that distract the reader. I've noted a few of instances of sentences that were difficult to read in the "technical comments" section, but I encourage the authors to carefully edit the full text.

AR: We will carefully revise the whole text regarding language and grammatical errors. Thank you for highlighting some examples.

Specific Comments –

R1C5: Page 2, Line 10: Why was vertically integrated water vapor transport analyzed and not column integrated water vapor? Since you are already investigating winds, it makes sense to investigate moisture without accounting for wind so that you can determine whether moisture availability is due to advection or local changes in temperature/evaporation? The two aren't completely independent, but they do have opposite seasonal cycles as well as their own sources of variability.

AR: We choose the atmospheric water transport instead of moisture content because we are analyzing dynamic precipitation controls, which have an effect on precipitation variability. They control when and where precipitation is higher or lower in specific regions. Moisture availability is a requirement for precipitation development.

R1C6: Page 2, Line 49-51: Referencing a study of deep tropical convection over the ocean hardly seems relevant to precipitation controls over the TP. It would be better to have a reference for mountain environments, or at least for land areas. If not, it may be necessary to show this relationship using HAR. Furthermore, this sentence is at odds with the negative effect of high horizontal wind speeds on convective precipitation that is described in following sentence.

AR: We agree that the paragraph is not well formulated which leads to the feeling that both sentences are at odds to each other. Horizontal wind speed has different impacts on precipitation depending on the atmospheric level and the type of precipitation. In lower atmospheric levels higher horizontal wind speeds can enhance moisture advection and also evaporation from the land surface. A feedback mechanism between wind, evaporation and convection exists. Higher horizontal wind speeds in higher atmospheric levels, for example the 300 hPa level above the Tibetan Plateau, have a distinct negative effect on convective precipitation by cutting off deep convection and wind shear effects. For a revised version of the manuscript we will revise the paragraph to make clear which effects can take place where and why. We will add references suitable for our study region.

R1C7: Page 3, Line 4: Note that the real benefit occurs because of orographic forcing of the moist flow. Without topography the relationship would be different.

AR: Thank you for this remark, we see that the sentence is not clearly enough formulated. We will revise the sentence and add the importance of the orographic forcing.

Old: "While the positive effect occurs in regions with mainly frontal/cyclonic precipitation because the moisture transport towards these regions is enhanced by the strengthened atmospheric flow."

New: "The positive effect occurs in regions with mainly frontal/cyclonic or orographic precipitation. Precipitation benefits from enhanced moisture transport by strengthened atmospheric flow, especially when the moisture flow is lifted up by orographic forcing. This plays an important role in our study region because of the high mountain ranges surrounding the Tibetan Plateau."

R1C8: Page 3, Line 15: It may be interesting to look at atmospheric stability (beyond the PBLH). In convective precipitation, vertical motions occur for different reasons than during frontal precipitation. Looking only at stability (theta or theta-e) would be instructive as to environmental conditions that support vertical motion irrespective of horizontal wind against the orographic barrier. This is important for both summer and winter. (this is only a suggestion)

AR: Thank you for this suggestion, but we think that this goes beyond the scope of the current study. Maybe it would be interesting to include analyses regarding the atmospheric stability in a subsequent study. In this study we focus on the variability of precipitation due to dynamic controls and not on its causes. Since this fact seems not to be clear for the reader we will emphasise the focus of the manuscript.

R1C9: Page 3, Line 41: This raises a number of questions for the reader: Were multiple PBLH schemes tested for WRF? What confidence is there that the PBLH from the configuration with the chosen parameterization is performing well? Were PBLH scheme sensitivity experiments performed in the development of HAR and if so, what observations are they validated with? If PBLH is to be included in this paper (which I've already argued against in the general comments section) it is necessary to address these very important questions in the manuscript rather than only providing the reference for HAR.

AR: As already mentioned earlier in the response, we will follow your suggestion, not to include PBLH as a precipitation control in a revised version of the manuscript.

R1C10: Page 4, Line 10: Because this study focuses on convective precipitation, some discussion of the validity of the HAR dataset, which uses a convective parameterization, should be included here. Beyond the reference, the authors should clearly note that precipitation during summer is highly sensitive to the choice of convective parameterization.

Furthermore, they should note that using higher resolutions and resolving precipitation explicitly may alter the spatial and temporal distribution of precipitation. Despite the issues associated with HAR precipitation, I do believe that it is very useful for understanding the mechanisms presented in this paper and the spatial and temporal differences in precipitation across large regions of the TP. I am simply encouraging the authors to further discuss the caveats here so that the reader is aware of these issues.

After all, they may not be familiar with the Maussion articles or may not read them to better understand the data that you have used.

AR: For a general description of the validation and evaluation of the HAR data please our response to reviewer comment R1C1. Of course using different parameterizations and higher spatial resolutions would change the precipitation values and its spatial and temporal distributions. But we assume that this would not change the the main results of this study since we use rank correlations which are independent of the mean values and scaling of the variables used in the correlations.

R1C11: Page 4, Line 32-33: I am curious as to how many days are included in the precipitation analysis. It seems that a threshold of 0.1 in the Karakoram and western Himalaya would remove very few days from consideration through the winter months. Why was a percentile-based threshold not used in addition to this low threshold to further exclude days with light precipitation that don't really contribute to seasonal totals in an area such as the KH. Including these dates masks the signal of precipitation controls during events that contribute strongly to overall precipitation accumulation (e.g. in the Karakoram and western Himalaya only a few dates during the winter season contribute the majority of seasonal (and in some cases, annual) precipitation).

AR: The threshold was basically used to filter out numerical artefacts in the data and not to exclude specific events from the data base. Since the precipitation distribution on Tibetan Plateau and surrounding high mountain ranges is very diverse, we do not want to exclude light precipitation events, because in some regions they can play a greater role for seasonal or annual precipitation than in the western Himalaya region. We decided to use a threshold that is able to filter out numerical artefacts but does not further reduce the number of precipitation days. We do not want to examine control mechanisms only for selected events. Maybe, we could examine whether the effect of selected dynamic variables on precipitation is different for extreme/intense precipitation events, in a subsequent study.

R1C12: Page 4 Line, 43-44: Do correlations in regions of sparse precipitation exhibit considerable sensitivity to a few extreme events? If you threshold out the largest few events (99th percentile) would the correlations change?

AR: I assume that if we would mask out the largest few events we would not have enough data for a reasonable correlation analysis. One of the conditions for a grid point to be considered for further analysis is a number of at least 13 precipitation days during all days of one month during the study period (2001-2013). Since the rank correlation is based on the ranks of the data and not on the values itself a few extreme events would just slightly change the results.

R1C13: Page 5, Line 20-22: The yellow and green classes may be very sensitive to the cumulus parameterization scheme that was used in HAR. Due to the uncertainty in how well WRF, and specifically HAR in this case, represent convective precipitation in the Himalaya, I don't think these classes can be so clearly defined. At the very least, better discussion of the caveats must be given, but it may be better to not distinguish these groups since the precipitation controls discussion does not.

AR: You are right that due to the use of a convective parameterization scheme uncertainties regarding the convective precipitation exist. But we think that there are

still differences in the impact of the monsoon system on these classes. We think it is an interesting result that these regions exhibit two different cluster. We explain in the text that it is not possible to draw a sharp border between the classes but that their existence can give a hint to the extent of the influence of the Indian summer monsoon on the precipitation on the Tibetan Plateau, a topic that is widely discussed in literature. Apart from choosing the number of clusters the cluster analysis itself is a quantitative and objective method. Since other numbers of clusters lead to similar results, the differentiation between these two clusters (yellow and green) is not an artefact. The results for varying the number of clusters are displayed in figure 5 in the response to reviewer comment R3C10.

R1C14: Page 6, Line 1: It seems to me that the reason the high elevations of the eastern Himalaya (near the Brahmaputra Channel) exhibit considerable precipitation contributions outside of the monsoon season is due to the orographic locking of westerly flow and the large amount of available moisture in this area. See Norris et al. (2015; their Figs. 5&6), which are focused on westerly disturbances affecting other regions of the Himalaya, but in both cases, exhibit precipitation in the eastern Himalaya generated by terrain locking of westerly flow.

AR: Thank you very much for this remark and the literature suggestion. We will consider this during the revision of the manuscript.

R1C15: Page 6, Line 30-32: This should cite some of the more seminal work on westerly disturbances.

AR: Please see our response to next comment.

R1C16: Page 6, Line 32-33: Higher wind speed does not necessarily mean enhanced moisture supply. There are many factors that modulate both wind and moisture in these systems (see Cannon et al. 2015). I would suggest just removing "which benefits from enhanced moisture due to higher wind speeds" and stating that the cyclonic/frontal precipitation is associated with westerly disturbances and then add a citation (e.g. Dimri et al. 2015 – review paper of westerly disturbances).

AR: Thank you very much for the literature suggestions. We will add some citations on westerly disturbances and a short description on the occurrence of westerly disturbances and their role for precipitation in the western part of our study region.

R1C17: Page 7, Line 2: Figure 1 indicates that the western Himalaya receives more than 0-5% of annual precipitation during summer (June alone is over 5% for Cluster 0). Furthermore, some studies argue that about 30%-60% of precipitation in this region falls during summer (e.g. Bookhagen and Burbank, 2010; their Fig. 4). Though the Bookhagen and Burbank estimate is probably too high because it is based on TRMM, which doesn't do well with winter precipitation, it is clear that considerable rainfall occurs in the western Himalaya during summer. Given that there is precipitation, what then could explain the lack of correlation?

The results show high positive correlations of precipitation with AWT in lower levels. The western disturbances which are responsible for the main part of precipitation in the western part of the study region occur primarily in winter. We assume that in summer the heating of the slopes in the western Himalayas leads to convection.

Therefore, moisture advection in lower levels seems to be more important than strong westerly winds at 300 hPa which act as a path way for westerly disturbances to the study region in winter. Our results are in good agreement with the precipitation amounts stated by Bookhagen and Burbank (2010). Cluster 0 receives ~35 % during May to October, while Cluster 5 receives ~60 % during the same time. The statement in the manuscript that there is almost no precipitation in the Pamir Karakoram region in summer is an error caused by changing the sentence from talking about July to the entire summer season without changing the values. We apologize for this error and for causing confusion. We will correct this in the revised manuscript. The value of 0-5% refers to Fig. 9 in Maussion et al. (2014) which shows the contribution of every month to the annual precipitation.

R1C18: Page 7, Line 7-8: Remove the linkage between higher wind speed and enhanced moisture. I would argue that higher wind speed more efficiently extracts moisture due to stronger orographic forcing. As mentioned before, there are many controls on moisture availability in the western Himalaya that are independent of the cross-barrier wind speed.

AR: The focus of the study is to analyse the effect of dynamic variables on the variability of precipitation and the mechanisms through which these controls can lead to an increase or decrease of precipitation. The variability of precipitation is determined by different controls according to different regions and seasons. Our results show that the atmospheric water transport and the wind speed in 300 hPa have different impacts on precipitation variability in different regions and seasons. Of course moisture availability is a limiting factor for precipitation development, but the causes of precipitation are not topic of this study. Since it seems that this was not clearly enough stated in the manuscript we will revise the manuscript to make this clear.

R1C19: Page 7, Line 32-39: The alternating positive/negative correlations look more like a gravity wave (i.e. updrafts and heavy precipitation on windward side of the mountain, downdrafts and light precipitation in the lee; Roe et al. 2005) than an error associated with aggregation. There should be an easy way to show this using topography aspect and wind direction, or using higher temporal resolution data available from HAR.

AR: We also think that it is a physically correct pattern, we just wanted to make the reader aware that we cannot exclude aggregation induced errors. Please see our response to reviewer comment R1C23. Although our study is quite extensive we cannot answer all questions exhaustively. We do not want to examine these questions more in detail here because they will be subject of upcoming research using the HAR data.

R1C20: Page 7, Line 41: I recommend looking at the correlation of precipitation and precipitable water rather than precipitation and water transport (or at least in addition to it). This is particularly important to do since convective precipitation requires enhanced moisture content, but not enhanced wind, while orographic precipitation requires both enhanced wind and enhanced moisture. Even though wind and moisture aren't independent, it would be interesting to see a figure of just the precipitable water correlations since it does not include the advection term, for which a similar variable was shown in previous figures (WS300, WS10). This could be particularly illustrative

since the seasonal cycles of moisture availability and wind speed are opposite for this region (e.g. Cannon et al. 2015).

AR: We will repeat the analysis for an additional variable representing the moisture content in the atmospheric column. Since we want to focus on dynamic precipitation controls we do not want to exclude the atmospheric water transport.

R1C21: Page 8, Line 31: I don't see the need to introduce boundary layer height, which is controlled by other processes that have already been evaluated. I don't think this adds much to the discussion, which was already quite strong. Furthermore, PBLH is parameterized in WRF, so you're introducing another question about how well this parameterization works, rather than focusing on variables that are explicitly resolved (moisture, temperature, wind). Though Maussion et al. 2011 performed a few sensitivity tests with different PBL parameterizations, validation specific to PBLH using radiosondes or wind profilers over the TP has never been done, so there are a lot of unknowns here. The lakes discussion is certainly interesting, but given the uncertainties about WRF's ability to resolve these processes, and because this discussion is not a main topic in the paper, I suggest leaving it out.

AR: We will follow your suggestion and will no longer include the PBLH analysis in a revised version of the manuscript. We still find the PBLH a very interesting variable, but think that your concerns are right. Since the PBLH is itself controlled by other variables it is probably not the best choice to analyze precipitation controls.

R1C22: Page 11, Line 33: The uncertainties in HAR should include the use of a convective parameterization, which directly affects the precipitation data (presence, timing and magnitude) that this study is based on. A better description of this particular issue is required since it is possible that some of the results in this work are sensitive to the choice of parameterization (or that if HAR were performed to explicitly resolve convection (<6km)).

AR: We will add a sentence about the use of a convective parameterization scheme and the related uncertainties to the section where we will discuss the validation/evaluation and uncertainties of the HAR. Please see our response to reviewer comment R1C1.

R1C23: Page 11, Line 48: Did you try using hourly or 3-hourly data to test what role aggregation errors play in your research? It seems that you have the necessary data to directly address this uncertainty.

AR: We also think that it is a physically correct pattern, we just wanted to make the reader aware that we cannot exclude aggregation induced errors. Maybe this remark was misleading the reader, so we will better explain our thoughts about the causes of this pattern. If an air flow hits a mountain range, the barrier causes orographic induced flow patterns, with updrafts on the windward side and downdraft on the lee side of the mountain range. This causes the precipitation to be smaller on average on the lee side because the downdrafts supress precipitation development. In the case of stronger flow to the mountains we also get stronger moisture advection and therefore the downdrafts are no longer able to supress precipitation. This leads to the simultaneous occurrence of precipitation and downdrafts which is the reason for the negative correlation patterns on the lee side of mountain ranges found in this study. To figure

out the role of aggregation errors goes far beyond the scope of this study. There are plans for future studies where the meteorological processes will be analysed explicitly using the available higher temporal resolutions. An aim of subsequent studies will be to assess the uncertainties of the dataset using, for example, newly available satellite data.

Technical Comments –

R1C24: Page 1, Line 7-8: Change "moisture" to "water resources"

AR: OK

R1C25: Page 1, Line 32: Change "strengthen" to "strengthening"

AR: OK

R1C26: Page 2, Line 1-8: A few of these sentences were difficult to read. A little restructuring would help the reader here.

AR: We will restructure these sentences in a revised version of the manuscript.

R1C27: Page 2, Line 25: When introducing the acronyms (PBLH, AWT and WS300), they should first appear next to their full names.

AR: OK

R1C28: Page 3, Line 7 & 18: "luv side" should be changed to "windward" as that is the common terminology in meteorology textbooks. (change in all instances)

AR: OK, thank you for this suggestion.

R1C29: Page 4, Line 24-25: Did you aggregate daily, or are you using a specific time-slice for each day?

AR: We aggregate the hourly model output to daily data. We use all time steps available for a day and not a specific time slice.

R1C30: Page 5, Line 21: Change "intensive" to "pronounced". Intensive makes it sound as though this relates to magnitude of precipitation values rather than the shape of the distribution.

AR: OK, done.

R1C31: Page 6, Line 13-14: remove "and therefore exhibit no coherent patterns". The patterns are coherent when considering mesoscale, synoptic and large-scale influences as well as topography. The patterns are complex and heterogeneous, but not incoherent.

AR: OK, thank you.

R1C32: Figure 2: The cyan and yellow lines are difficult to see. Perhaps all the lines could be thicker or darker colors could be used

AR: For a revised version of the manuscript we will also revise this figure to make it easier to read.

R1C33: Page 7, Line 12: "There are high positive correlations (over the Tibetan Plateau?) between WS10 and precipitation. . ."

AR: Yes. We will add the region to the sentence.

R1C34: There are too many small mistakes in word choice, grammar and sentence structure to identify each individually. I encourage the authors to carefully edit the full text.

AR: We will carefully revise the manuscript regarding word choice, grammar, and sentence structure.

Anonymous Referee #2

This manuscript analyzed the seasonality and spatial variability of dynamic precipitation controls on the Tibetan Plateau using HAR dataset. It stresses the high impact of the mid-latitude westerlies on precipitation distribution on the TP and its surrounding year-round. I can feel the strong eagerness of authors on concluding the westerly is the controller of the precipitation over the TP. However, manuscript is supportive enough. I have great concerns before the conclusions could be drawn. Substantial revisions are necessary before the manuscript is publishable.

R2C1: All of this work is based on the single approach – correlation and single data set – HAR. Large uncertainties in conclusions occur due to the approach adapted and data set used. Multiple approaches or datasets are necessary to swipe away these uncertainties.

AR: The HAR data set is physically based and also forced with a physically based data set. The comparison with ERA-Interim and the validation with TRMM and station data show that the HAR is suitable to analyse precipitation variability. It is true that we cannot remove all uncertainties. But since the HAR data set is the only dataset with such a high spatial and temporal resolution available for the entire Tibetan Plateau and surrounding high mountain ranges we do not have a real opportunity to use multiple datasets. We will add a section about the uncertainties of the HAR data and describe the validation and evaluation done for the data so far.

R2C2: More validation and evaluation on HAR are of fundamental necessity before it could be used on analyzing.

AR: Please see our response to reviewer comment R1C1. In a revised version of the manuscript we will add a paragraph about the validation and evaluation of the HAR data and the resulting uncertainties.

R2C3: The whole basement of this study is the precipitation classification by Maussion et al. (2014). However, the precipitation classification Maussion et al (2014) did is for only the glacier accumulation regimes locating at high altitudes above about 5000m shown in their Fig. 14, rather than for the whole Tibetan. Is it representative for the precipitation over the whole TP? If so, please show the evidences. If not, suggest changing the title to ". . .on the glacier accumulation regimes over the Tibetan Plateau"

AR: The precipitation classification in this study is based on the idea to use a clustering method to determine accumulation/precipitation regimes for glacier regions done by Maussish et al. (2014). We conducted a cluster analysis for the entire Tibetan Plateau and surrounding high mountain ranges using the HAR data, which should be clear if looking at figure 2. Since this seems not to be clear we will revise this section. Since we do not focus on specific areas, e.g. glacier areas, our precipitation classification is based on a larger data set and provides more information.

R2C4: P2L47, it reads "on average, more than 60% of moisture needed for precipitation falling on the inner TP are provided by the TP itself (Curio et al. 2015)". In authors' previous paper published in 2015. That suggests that convections over the TP dominate precipitation in the TP rather than the moisture transportation from outside. In this manuscript, the westerly are argued to be the dominant controller in precipitation. These two conclusions are conflict to each other. Which one is the leading controller of the precipitation in the TP, in authors' ultimate view? What is the linkage of the convections and the westerly? Considerate analysis and evaluation are strongly suggested before the conclusion is drawn.

AR: We do not think that these two findings are conflict to each other. The moisture recycling is the dominant feature regarding the question where the moisture for precipitation on the TP is coming from, while the dynamic precipitation controls related to the subtropical jet (wind speed at the 300 hPa level and the atmospheric water transport) are found to be dominant regarding the variability of the precipitation. One is the provider of necessary moisture while the others are dynamic effects which control precipitation variability. The subtropical jet causes a decrease of precipitation in regions which are dominated by convective precipitation while it causes an increase of precipitation in regions which are dominated by frontal/orographic precipitation.

R2C5: Over ocean or area with low elevations, 300 hPa is high enough to stand for the height that the westerly locates. However, the TP possesses an elevation above 4000m on average. 300 hPa is too low for the westerly over there. Authors cite Schiemann et al. (2009) as the reason of this selection. We can read the cited reference is about the precipitation climate of Central Asia. The TP possesses distinguish climate from the Central Asia not only in its unique height, but also the distance from the ocean. They are not comparable. Authors should refer to works in the TP rather than other where. Numerous studies claim that the westerly reaches as high as 100 hPa over the TP. For instance, 200 hPa (in the global climate model domain, Gao et al., J. Climate 2014) or 100 hPa (in regional climate model domain, Gao et al., J. Climate 2015) are the height where the westerly hang over the Tibetan; whereas, the 600 hPa (in the global climate model domain) or 500hPa (in regional climate model domain) is the near surface. The 300 hPa is a middle layer between the upper and near surface layers. It is reasonable using the vertical wind speed at 300 hPa for the vertical motions. However, the horizontal wind speed at 300 hPa used to represent the westerly jet over the Tibetan is questionable.

AR: We decided to use the wind speed at the 300 hPa level because the wind shear in this height more strongly suppresses deep convection than at the 200 hPa level where the core of the jet lays. Also we assumed that the results would not change overall using the wind speed in 200 hPa. To proof that, we repeated the correlation analysis between wind speed and precipitation for the wind speed in 200 hPa (figure R3). The results are very similar. The correlations at the 300 hPa level (figure R4) are slightly higher because the negative effect of higher wind speeds on precipitation by cutting off deep convection is higher at this level than in 200 hPa. We will add this analysis to the supplement of the revised manuscript.

Another supporting reasons is the finding by Mölg et al. (2014) that the flow strength at the 300 hPa level has a strong influence on precipitation in the regions influenced by the monsoon. Furthermore, the westerly jet reaches nearly down to the surface over the Tibetan Plateau and the 300hPa level lays still within the band of westerly winds with higher wind speeds associated with the jet stream as shown by Maussion et al. (2014) in their figure 2.

[Figure]

Fig. R3: Coefficient of correlation (rho) between the horizontal wind speed at 200 hPa and precipitation for all months (01-12). Positive correlations are denoted in red, while negative correlations are denoted in blue. Only correlations significant at the 0.05 significance level are shown.

[Figure]

Fig. R4: Coefficient of correlation (rho) between the horizontal wind speed at 300 hPa and precipitation for all months (01-12). Positive correlations are denoted in red, while negative correlations are denoted in blue. Only correlations significant at the 0.05 significance level are shown (originally figure S3 of the supplement).

R2C6: It is claimed that "Six controllers are selected". However, five are analyzed in 3.2.1a, 3.2.1b, 3.2.2, 3.2.3 and 3.2.4. Do I missing something? Before the controller is concluded, I prefer to call them elements rather than controller. In addition, background of these elements is missing. Why these elements are chosen? What is their relevance with precipitation? For instance, the horizontal wind speed at model level 10 (WS10) is used. What is the height of model level 10? What it stands for? Is the PBLH relevant to PBL parameterization schemes used in simulation?

AR: You are right that we only describe five precipitation controls in the section about the correlations with precipitation. We left out the vertical wind speed at model level 10, because we did not want to repeat things which are quite similar to each other. In a revised version of the manuscript we will remove the analysis of the vertical wind speed at model level 10. Since we will also no longer use PBLH as a precipitation control (due to reviewer suggestions) we will only analyse four variables as precipitation controls in the revised version of the manuscript.

The height of model level 10 is about 2 km above ground over the Tibetan Plateau. We chose this level to represent processes taking place within the boundary layer, which can exceed 3 to 4 km above ground over the Tibetan Plateau. It is located in the middle to upper range of the planetary boundary layer.

We will consider the use of the term "elements" instead of "controls". But maybe we can keep the term controls if we better explain that precipitation controls do not necessarily control whether precipitation occurs or not, but rather (control) the spatial and temporal variability of precipitation.

We describe each element in the introduction and their relevance for precipitation. Since it seems not to be clear enough we will revise this section. We made the mistake not to state clearly enough that the selected elements control the precipitation variability. We just named precipitation development which leads the reader to the wrong assumption that we claim that there would be no rain without the influence of the dynamic precipitation controls. To make this clear will be part of the revision of the manuscript.

Anonymous Referee #3

Summary:

The study described in this manuscript investigates the impact of five selected dynamic controls (horizontal wind speeds at different levels, vertical wind speed, atmospheric water transport and planetary boundary layer height) for precipitation in High Asia. The study's novelty lies in the use of a relatively new high resolution dataset to answer questions that have previously not been addressed. The conclusions reached are supported by the findings. They are scientifically valuable and presented in a logical and intuitive way. The title and abstract are concise. However, I do have a few questions, concerns and suggestions regarding the methods applied and language used in the manuscript. After consideration of those and moderate revisions I recommend publication of this manuscript.

General Comments:

R3C1: The statistical methods used in this study are suitable for addressing the questions. However, since the findings and conclusions rely heavily on purely statistical methods (mostly Spearman's rank correlation), I would like to see them discussed in more detail (see specific comments). The authors should also describe how the significance of rho was calculated. (I believe this is missing entirely from the manuscript). The PCA is suitable for describing the spatio-temporal variability of correlation of controlling factors and precipitation.

AR: Please see response to comment R3C8 for a detailed description of the statistical method and the calculation of the significance.

R3C2: I am not sufficiently familiar with specific shortcomings of the HAR dataset to comment on whether or not the authors adequately addressed these in the manuscript and took them into consideration when drawing conclusions. However, since this study is entirely based on HAR, I would appreciate some comments on existing uncertainties of the dataset and how/whether or not this limits the interpretation of the results of this study.

AR: Please the response to comment R1C1 regarding the same topic.

R3C3: There are numerous grammatical errors throughout the manuscript and sentence structure is often confusing. This makes it very difficult to read and understand in certain sections. I highlighted some of these in the "technical comments" below. I strongly recommend the authors edit the language of the manuscript and let a native speaker (or someone with a similar level of written English) review it before resubmission.

AR: Thank you for this remark and recommendation. We will carefully revise the manuscript, remove grammatical errors and improve the sentence structure.

Specific Comments:

R3C4: Page 2, line 8: You write here and later on that you select six factors. However, you only list five here (and in the results). Maybe I am missing something?

AR: You are right that we only describe five precipitation controls in the section about the correlations with precipitation. We left out the vertical wind speed at model level 10, because we did not want to repeat things which are quite similar to each other. In a revised version of the manuscript we will show and discuss the figures and the differences to the correlations of the wind speed in 300 hPa and precipitation.

R3C5: Page 2, line 15: What is this assumption based on? I expected a little more explanation for the selection of controlling factors – scientific or purely technical. Also, what factors were excluded and why?

AR: Since the precipitation distribution on the Tibetan Plateau is influenced by both the Indian Summer Monsoon and the mid-latitude westerlies (e.g. Maussion et al., 2014; Böhner et al., 2006), we focus on dynamic precipitation controls which are related to these two atmospheric circulation features. We have to make more clear that the focus of the study is not to examine precipitation development but the reasons for precipitation variability. We apologize that the former use of the phrase precipitation development was misleading the reader. This variability is influenced by the atmospheric circulation and the related variables like horizontal and vertical wind and moisture transport. We choose variables from the HAR data set, which are already proven to be in good agreement with observations and gridded datasets. The selected controls are well known to have an impact on precipitation variability. The positive effect of AWT on precipitation variability is described in, for example, Barros et al. (2006), Giovannetone et al. (2009) and Zhou et al. (2005). The positive effect of higher wind speeds on moisture advection and orographic lifting, and in lower levels enhancing evaporation and therefore their positive correlations with precipitation are known in the literature (e.g. Johansson and Chen (2003), McVicar (2012), Roe (2005)). Higher wind speeds in higher atmospheric levels can cut of deep convection via the wind shear effect, studies showing this effect are, for example, Findell et al. (2003) and Zhang and Atkinson (1995). Rose et al. (2003) show effects of vertical wind speed and most important its direction on precipitation. Updrafts lead to an increase of precipitation while downdrafts lead to a decrease of precipitation. Of course there are other variables influencing precipitation, but we think that our selection is suitable for our study region. We will add some of the references to the introduction of the revised manuscript to support our selection of variables.

R3C6: Page 4, line 33: How was the 0.1mm threshold chosen? In context of the study's aims, what are the advantages of choosing an absolute value instead of a grid-box specific percentile for example?

AR: The threshold was basically used to filter out numerical artefacts in the data and not to exclude specific events from the data base. The threshold of 0.1 mm per day is also a commonly used minimum value to define a precipitation day (e.g. Martin-Vide, J., & Gomez, L. (1999), Ceballos et al. (2004), Polade et al. (2014), Lana et al. (2006), Liu et al. (2011), Bartholy et al. (2010), Frei et al. (1998)).

R3C7: Page 4, line 37: It may be more insightful to highlight the advantages of the Spearman rank correlation over other measures of statistical dependence for this particular question involving precipitation and its controlling factors.

AR: We will add a description of the statistical methods in the revised version of the manuscript. The Spearman rank correlation was chosen because this measure is more robust against outliers in the data. Since the spatial and temporal distribution of precipitation is very different in specific region on the Tibetan Plateau the range of values is very large. The rank correlation also can help to reduce the effect of very intense or very light rain events on the correlation results.

R3C8: Page 4, line 39: There are multiple ways in which the statistical significance of such a correlation can be determined. I recommend that you at least mention in one or two concise sentences how it was determined in this study. Also, how sensitive are your results to different, commonly used significance levels, e.g. 0.01? Since the correlation analyses form the centre piece of your study (and the PCA's are also based on correlation coefficients), I think it is necessary to provide a little more insight.

AR: The statistical significance of the results was tested using a two tailed test to determine the deviation from zero. We used the r_correlate routine of the software IDL to compute the Spearman rank correlation and its level of significance. The source code references to the textbook "Numerical Recipes, The Art of Scientific Computing (Second Edition), Cambridge University Press, ISBN 0-521-43108-5". The pages 640-642 describe the Spearman Rank-Order Correlation Coefficient. The two-sided significance level is extracted from the t-value based on the degree of freedom (N-2) and its beta distribution. The routine returns the variable probrs, which is the p-value and gives the significance of the correlation coefficient. A small value of probrs indicates a significant correlation. This value has to be compared with the value set for the significance level in order to determine whether it is significant at the given significance level or not.

rs: Spearman rank-order correlation coefficient

N-2: degrees of freedom

t = rs * sqrt(N-2 / 1-rs2)                              ;its t-value

probrs = betai(0.5 * (N-2), 0.5, (N-2) / (N-2) + t * t)      ;its significance, p-value

To test, how sensitive the results are regarding the use of a different significance levels, we repeated the analysis with the significance level 0.01 for the correlation between

wind speed in 300 hPa and precipitation (figure R5) and compared the results with the results gained using 0.05 as significance level (figure R6). The resulting patterns do not change very much. The areas with negative and positive correlations are a little bit smaller, but the changes occur where the values are already lower (at the borders of the correlation patterns). The regions with the highest correlations stay stable and even small areas of positive or negative correlations do not disappear.

[Figure]

Fig. R5: Coefficient of correlation (rho) between the horizontal wind speed at 300 hPa and precipitation for all months (01-12). Positive correlations are denoted in red, while negative correlations are denoted in blue. Only correlations significant at the 0.01 significance level are shown.

[Figure]

Fig. R6: Same as Fig. R5, but only correlations significant at the 0.05 significance level ar shown (originally figure S3 of the supplement).

R3C9: Page 5, line 1-3: Maussion et al. use this clustering approach to classify glacier accumulation regimes. While the analysis itself is a universal tool of of descriptive statistics fitting for the aims of the study, I recommend the addition of justification for deviation from the Maussion et al. clustering since you present this as something that builds on that study. For example, why choose seven instead of five clusters as Maussion et al. do? (see comment below)

AR: Please see response on next comment.

R3C10: Page 5, line 5-8: As I understand, you varied k for the clustering to determined the optimal number, i.e. the number giving you good coherence within the classes and sufficient distinctions between them. What was the k range you used and how did you determine optimal k? Was this apparent from a qualitative assessment of plotted results or was it determined by something like a discriminant analysis? Furthermore, for similar clusters such as purple/red and yellow/green, I recommend adding comments on the sensitivity of the clustering to physical conditions versus HAR/WRF limitations. In other words: is the separation of yellow and green physically meaningful?

AR: We varied the k for clustering from 5 to 9, we choose to start with 5 because this was the k used in the Maussion et al. (2014) study for glacier regions. We found 7 to be the optimal k for the recent study since it gave us good coherence within the classes and sufficient distinctions between them, like you say and we state in the manuscript. We determined this qualitatively by looking at the plots for the different numbers of

clusters. We first conducted the cluster analysis with 5 clusters like Maussion et al. (2014). But since we included a much higher number of grid points (we analysed the entire Tibetan Plateau), the results for the areas included in both analyses look slightly different especially in the Karakoram and Tien Shan. By increasing the number of cluster to 6, one cluster that covers most parts of the northern part of the study region, northern Tibetan Plateau and Tarim Basin, breaks up in two cluster. Setting the number to 7, we get more variation in the Karakoram and Tien Shan which looks more similar to the cluster distribution achieved by Maussion et al. (2014). Using a higher number of clusters led to the occurrence of more clusters of smaller size which are not as distinct from each other as the larger ones. In general, it would be interesting to have more cluster to get a higher spatial differentiation. But by increasing the cluster number the spatial coherence decreases and therefore the interpretability also decreases. We decided to use seven clusters as a compromise between higher spatial differentiation and less spatial coherence. If requested, we can add the plots for the different cluster numbers to the supplement of the manuscript to give the reader the chance to get an idea of the differences. Since the core areas of the clusters stay stable while changing the number of clusters we do not think that the clusters are an artefact. The results for the different cluster numbers are shown in figure R7.

Add figures for varying number of clusters!!!

R3C11: Page 7, line 32: Could this pattern be the result of a gravity wave?

AR: Thank you very much for this remark. It is possible that this pattern could be related to a gravity wave. But since this pattern only shows the correlation between wind speed and precipitation I am not sure if it is straight forward to interpret the alternating pattern between positive and negative correlations as a gravity wave pattern. The existence of high mountain ranges causes up- and down-drafts and gravity waves are likely to occur when atmospheric flow passes high mountain ranges. But gravity waves require stable atmospheric conditions under which mostly no precipitation occurs. Since we focus on precipitation days these situations are not captured in the data. Since we are looking at the mean daily conditions for precipitation days for all days of a month during our study period of 13 years, we assume that a pattern caused by gravity waves in combination with precipitation would be averaged out.

R3C12: Page 12, line 1-2: This is a very general comment and does not go into the type of correlation you are dealing with in this study. I believe Spearman's R to be adequate here, but I recommend adding a word of caution, since a nonparametric measure for correlation limits the interpretation of R values. It is not a source of uncertainty as such, but should be mentioned somewhere in the manuscript (along with more details on the significance tests and the sensitivity of results to significance levels)

AR: Thank you for this remark. We are aware of the fact that ranking the data leads to a slight information loss compared to the real values. But we are sure that for our study the advantages are bigger than the disadvantages. Rank correlations are more robust against outliers and are independent from the real data. In our case it makes it easier to compare regions with very different precipitation amounts.

Technical Comments:

R3C13: There are too many grammatical errors to highlight all in this section. Furthermore, the phrasing of many sentences is confusing. I strongly suggest to let a native speaker to review the language of this manuscript.

AR: Thank you for this comment. We will carefully revise the language of the manuscript with help of a native speaker.

R3C14: Page 1, line 33: "strenghtening" instead of "strengthen"

AR: OK, done.

R3C15: Page 1, line 31-33: This sentence is confusing. Think about rephrasing it or breaking it up into two sentences.

AR: We will break up this sentence into two sentences.
Old: "This may help to estimate the impact of future climate change on precipitation development and amounts, since if a specific control in a specific region and time is identified, one would be aware that a strengthen or weakening of this precipitation control has distinct impact on precipitation development."

New: "Identifying precipitation controls may also help to estimate the impact of future climate change on precipitation variability."

R3C16: Page 2, line 3: Do you mean "gives us the opportunity for a process based analysis of the data"?

AR: Yes, thank you.

R3C17: Page 2, line 5-7: Change to something like "Therefore, we want to examine the timing, location and strength of the influence of precipitation controls on precipitation development."

AR: OK, done. Thank you for your suggestion.

R3C18: Page 2, line 7-8: Rephrase to something like "The aim of this study is to describe the spatial and temporal correlation of [...]"

AR: Done. New: "The aim of this study is to describe the spatial and temporal correlation of selected dynamical variables and precipitation. We want to reveal the underlying mechanisms through which the variables influence precipitation variability and therefore act as controls of precipitation variability."

R3C19: Page 2, line 12: I suggest citations at this point when making statements about the influence of the factors being known.

AR: We will add citations in a revised version of the manuscript.

R3C20: Page 2, line 25: Abbreviations should be introduced earlier in the text when their full names are first mentioned.

AR: OK, sorry, this was a mistake maybe caused by changing the order of paragraphs.

R3C21: Page 2, line 33: Change to "correspond to"

AR: OK, done.

R3C22: Page 2, line 35: Change to "at a height"

AR: OK, done.

R3C23: Page 2, line 36: "[. . .] which strength and location [. . .]" - review the grammar here/rephrase.

AR:
Old: At the 300 hPa level we are already in a height where the westerly jet occurs, which strength and location influences the hydro-climate of the TP and central Asia (e.g. Schiemann et al., 2009).

New: The core of the westerly jet occurs at the 200 hPa level. Over the Tibetan Plateau the jet reaches down to 300 hPa and still has an effect there and also in lower levels. This was shown for the HAR by Maussion et al. (2014). The strength and location of the jet influences the hydro-climate of the Tibetan Plateau and central Asia (e.g. Schiemann et al., 2009).

R3C24: Page 3, line 7: "windward side" is more commonly used in English than "luv side" (which is still common in German literature). I recommend changing it throughout the manuscript.

AR: Thank you for this advice, we will change it.

R3C25: Page 3, line 45-47: This is not gramatically sound (see comment Page 2, line 12)

AR: We changed the sentence, see below please.
Old: i. analyse the impact of the selected dynamic controls on precipitation development spatially and temporally differentiated, where and when has which precipitation control a how strong influence,

New: i. analyse the impact of selected dynamic controls on the spatial and temporal variability of precipitation.

R3C26: Page 4, line 1-2: This is not gramatically correct.

AR: We changed the sentence, see below please.
Old: ii. examine if precipitation controls have always and everywhere in the study region the same impact or whether there are opposing effects active in different sub-regions,

New: ii. examine whether the controls act in the same way in different regions and at

different times,

R3C27: Page 5, line 19: Correct to "[. . .] precipitation is falling in this region, [...]"

AR: OK, done.

R3C28: Page 5, line 25: "There" should be the start of a new sentence for this to be grammatically sound.

AR: Thank you for this comment, please see changes below.
Old: The class laying between the so called monsoonal and convective classes represents a region of mixing between monsoonal and convective classes, there we can have monsoonal precipitation and/or only solar forced convective precipitation (green). This class represents a transition zone where the monsoon can have influence but is not dominant.

New: The green class represents a transition zone between the monsoonal and convective classes where the monsoon can have an influence but is not dominant. This means that both monsoonal precipitation and/or only solar forced convective precipitation can occur.

We will also rephrase the section where the precipitation clusters are introduced to make clearer what the names of the classes mean and what makes the classes different from each other. Of course convection is dominant in the monsoonal precipitation class, but the timing of precipitation is different to that in the convective class.

R3C29: Page 5, line 30-33: This sentence is confusing and not grammatically correct. Please rephrase.

AR: Thank you for this comment, please see changes below.
Old: That there is a forcing difference between the two classes is clearly visible in the timing and strength of the precipitation maximum which is higher and later in the monsoonal class and starts during the Indian Summer Monsoon onset period and persists only for a shorter time period.

New: The timing and strength of the precipitation maximum provides an indication for different forcing. In the monsoonal class the precipitation maximum is higher, starts during the Indian Summer Monsoon onset, and persists for a shorter time period.

R3C30: Page 5, line 35: Correct to "its"

AR: OK, done.

R3C31: Page 5, line 44: Correct to "evenly" (adverb)

AR: OK, done.

R3C32: Page 7, line 30: Do you mean "This supports the interpretation [...]"?

AR: Yes. We will rephrase the sentence.

R3C33: Page 8, line 4: Correct to "Therefore"

AR: OK, done.

Anonymous Referee #4

This paper by Julia Curio and Dieter Scherer try to demonstrate that the westerlies is an important factor influence the precipitation in the Tibetan Plateau even in the monsoon season, based on the High Asia Refined analysis. The correlation and principal components are used in this manuscript. There are five major problems in the manuscript described below. By considering these issues, I suggest major revisions.

Major issues:

R4C1: The text is not fairly organized, especially the introduction. The discussion and interpretations are superficial given the figures results. Authors need to present their work with better clarity.

AR: We will revise the entire manuscript to make it more concise, clear and easier to follow for the reader. We also will try to present our findings more clearly and to deepen the discussion.

R4C2: Authors used the HAR as the unique data for analyses, but they did not evaluate the dataset with observations on the Tibetan Plateau, and did not confirm their results with other data. The systematic analysis of stable isotopes in precipitation on the Tibetan Plateau has demonstrated the seasonal moisture origins and moisture transports. I suggest the authors to refer it.

AR: We will add a section about the uncertainties of the HAR data set and the evaluation /validation. In our last study about atmospheric moisture transport on and to the Tibetan Plateau we referred to studies analyzing the stable isotopes in precipitation to identify moisture sources and transport routes. Since this is not the topic of the current study we only referred to our former findings. In a revised version of the manuscript we will refer to these analyses to support the importance of moisture recycling on the Tibetan Plateau.

R4C3: Why these six factors are considered in this study? Winds does not mean the precipitation if there is no moisture transport with them. Authors concluded that all of these factors combined influence on precipitation. In this case, what is the main contribution of this study? Authors also said moisture recycling was important that offers more than 60% moisture for precipitation. Which one is more important, the westerlies or the recycling? It is unclear.

AR: We do not think that these two findings should be ranked. One finding is regarding the moisture sources the other regarding the dynamical processes that lead to precipitation variability.

The horizontal wind speed is not described as a factor which alone leads to precipitation. The selected controls do not cause or suppress precipitation but influence the spatial and temporal variability of precipitation. The main influence of the horizontal wind speed is cutting off deep convection, if it is not associated with enhanced moisture transport.

R4C4: Why is 300hPa for the westerlies? Authors did not clarify the precipitation heights on the Tibetan Plateau. Due to the complex topography and climate, different type of precipitation shows diversified contribution to the annual precipitation amount and it occurs at different height. It should be considered.

AR: Please see response to reviewer comment R2C5.

R4C5: This manuscript is not easy to follow because of many grammatical errors and disordered sentences.

AR: We will improve the language and sentence structure in a revised version of the manuscript.

References

Barros, A. P., Chiao, S., Lang, T. J., Burbank, D., & Putkonen, J. (2006). From weather to climate—Seasonal and interannual variability of storms and implications for erosion processes in the Himalaya. *Geological Society of America Special Papers*, *398*(02), 17 –38. doi:10.1130/2006.2398(02).

Bartholy, J., & Pongracz, R. (2010). Analysis of precipitation conditions for the Carpathian Basin based on extreme indices in the 20th century and climate simulations for 2050 and 2100. *Physics and Chemistry of the Earth*, *35*(1-2), 43–51. doi:10.1016/j.pce.2010.03.011

Bookhagen, B., & Burbank, D. W. (2010). Toward a complete Himalayan hydrological budget: Spatiotemporal distribution of snowmelt and rainfall and their impact on river discharge. *Journal of Geophysical Research*, *115*(F3), F03019. doi:10.1029/2009JF001426

Ceballos, A., Martinez-Fernandez, J., & Luengo-Ugidos, M. A. (2004). Analysis of rainfall trends and dry periods on a pluviometric gradient representative of Mediterranean climate in the Duero Basin, Spain. *Journal of Arid Environments*, *58*(2), 215–233. doi:10.1016/j.jaridenv.2003.07.002

Curio, J., Maussion, F., & Scherer, D. (2015). A 12-year high-resolution climatology of atmospheric water transport over the Tibetan Plateau. *Earth System Dynamics*, *6*(1), 109–124. doi:10.5194/esd-6-109-2015

Findell, K. L. (2003). Atmospheric controls on soil moisture-boundary layer interactions: Three-dimensional wind effects. *Journal of Geophysical Research*, *108*, 1–21. doi:10.1029/2001JD001515

Frei, C., Schär, C., Lüthi, D., & Davies, H. C. (1998). Heavy precipitation processes in a warmer climate. *Geophysical Research Letters*, *25*(9), 1431. doi:10.1029/98GL51099

Garreaud, R. D. (2007). Precipitation and circulation covariability in the extratropics. *Journal of Climate*, *20*(18), 4789–4797. doi:10.1175/JCLI4257.1

Giovannettone, J. P., & Barros, A. P. (2009). Probing Regional Orographic Controls of Precipitation and Cloudiness in the Central Andes Using Satellite Data. *Journal of Hydrometeorology*, *10*(1), 167–182. doi:10.1175/2008JHM973.1

Johansson, B., & Chen, D. (2003). The influence of wind and topography on precipitation distribution in Sweden: Statistical analysis and modelling. *International Journal of Climatology*. doi:10.1002/joc.951

Lana, X., Burgueño, A., Martínez, M. D., & Serra, C. (2006). Statistical distributions and sampling strategies for the analysis of extreme dry spells in Catalonia (NE Spain). *Journal of Hydrology*, *324*(1-4), 94–114. doi:10.1016/j.jhydrol.2005.09.013

Liu, B., Xu, M., & Henderson, M. (2011). Where have all the showers gone? Regional declines in light precipitation events in China, 1960-2000. *International Journal of Climatology*, *31*(8), 1177–1191. doi:10.1002/joc.2144

Martin-Vide, J., & Gomez, L. (1999). Regionalization of peninsular Spain based on the length of dry spells. *International Journal of Climatology*, *19*(5), 537–555. doi:10.1002/(SICI)1097-0088(199904)19:5<537::AID-JOC371>3.0.CO;2-X

Maussion, F., Scherer, D., Mölg, T., Collier, E., Curio, J., & Finkelnburg, R. (2014). Precipitation Seasonality and Variability over the Tibetan Plateau as Resolved by the High Asia Reanalysis*. *Journal of Climate*, *27*(5), 1910–1927. doi:10.1175/JCLI-D-13-00282.1

McVicar, T. R., Roderick, M. L., Donohue, R. J., Li, L. T., Van Niel, T. G., Thomas, A., … Dinpashoh, Y. (2012). Global review and synthesis of trends in observed terrestrial near-surface wind speeds: Implications for evaporation. *Journal of Hydrology*, *416-417*, 182–205. doi:10.1016/j.jhydrol.2011.10.024

Mölg, T., Maussion, F., & Scherer, D. (2014). Mid-latitude westerlies as a driver of glacier variability in monsoonal High Asia. *Nature Climate Change*, *4*(1), 68–73. doi:10.1038/nclimate2055

Numerical Recipes, The Art of Scientific Computing (Second Edition), Cambridge University Press (ISBN 0-521-43108-5).

Polade, S. D., Pierce, D. W., Cayan, D. R., Gershunov, A., & Dettinger, M. D. (2014). The key role of dry days in changing regional climate and precipitation regimes. *Scientific Reports*, *4*(4364), 8pp. doi:10.1038/srep04364

Rose, B. E. J., & Lin, C. A. (2003). Precipitation from vertical motion: A statistical diagnostic scheme. *International Journal of Climatology*, *23*(8), 903–919. doi:10.1002/joc.919

Zhang, J. W., & Atkinson, B. W. (1995). Stability and wind shear effects on meso-scale cellular convection. *Boundary-Layer Meteorology*, *75*(3), 263–285. doi:10.1007/BF00712697